# Replacing conventional battery electrolyte additives with dioxolone derivatives for high-energy-density lithium-ion batteries

Sewon Park[1], Seo Yeong Jeong[2], Tae Kyung Lee[1,3], Min Woo Park[1], Hyeong Yong Lim[1], Jaekyung Sung[1], Jaephil Cho[1], Sang Kyu Kwak [1✉], Sung You Hong [2✉] & Nam-Soon Choi [1✉]

Solid electrolyte interphases generated using electrolyte additives are key for anode-electrolyte interactions and for enhancing the lithium-ion battery lifespan. Classical solid electrolyte interphase additives, such as vinylene carbonate and fluoroethylene carbonate, have limited potential for simultaneously achieving a long lifespan and fast chargeability in high-energy-density lithium-ion batteries (LIBs). Here we report a next-generation synthetic additive approach that allows to form a highly stable electrode-electrolyte interface architecture from fluorinated and silylated electrolyte additives; it endures the lithiation-induced volume expansion of Si-embedded anodes and provides ion channels for facile Li-ion transport while protecting the Ni-rich $LiNi_{0.8}Co_{0.1}Mn_{0.1}O_2$ cathodes. The retrosynthetically designed solid electrolyte interphase-forming additives, 5-methyl-4-((trifluoromethoxy)methyl)-1,3-dioxol-2-one and 5-methyl-4-((trimethylsilyloxy)methyl)-1,3-dioxol-2-one, provide spatial flexibility to the vinylene carbonate-derived solid electrolyte interphase via polymeric propagation with the vinyl group of vinylene carbonate. The interface architecture from the synthesized vinylene carbonate-type additive enables high-energy-density LIBs with 81.5% capacity retention after 400 cycles at 1 C and fast charging capability (1.9% capacity fading after 100 cycles at 3 C).

[1] School of Energy and Chemical Engineering, Ulsan National Institute of Science and Technology (UNIST), Ulsan, Republic of Korea. [2] Department of Chemistry, Ulsan National Institute of Science and Technology (UNIST), Ulsan, Republic of Korea. [3] Photovoltaics Research Department, Korea Institute of Energy Research (KIER), Daejeon, Republic of Korea. ✉email: skkwak@unist.ac.kr; syhong@unist.ac.kr; nschoi@unist.ac.kr

Lithium-ion batteries (LIBs) have been unrivaled energy sources for portable devices, such as laptops and smartphones, over the last three decades. The materials technology and the manufacturing processes for LIBs have advanced considerably, which have vastly improved their capacities and rendered them capable of powering electric vehicles (EVs)[1–5]. Securing high-energy-density LIBs with a long lifespan and fast charging performance is vital for realizing their ubiquitous use as superior power sources for electric vehicles. Among the materials developed for EV-adoptable high-energy-density LIBs, Si, and Ni-rich layered oxides have been prime choices for electrode material construction, owing to their high-energy storage capabilities[6–10]. However, Si-based anodes and Ni-rich cathodes suffer from structural instabilities induced by anisotropic volume changes and interface deterioration. Unlike graphite, the lithiation of Si provokes the generation of Li–Si alloys, which cause a colossal volume expansion (>300%) and fatal mechanical fractures of the Si particles[10,11]. Therefore, the solid electrolyte interphases (SEIs) at Si anodes degrade severely. This degradation induces the exposure of the Si surface, which leads to the continuous electrolyte decomposition-induced thickening of the SEI and eventual electrolyte depletion, thus rendering the battery unusable[12].

Electrolyte additives have been extensively employed for extending the cycle life of LIBs while preventing electrolyte decomposition at the electrodes[13–17]. So far, reductive compounds possessing fluorine-donating moiety or vinyl group[18–21] have been exploited as SEI-forming additives for Si-based anodes. Fluoroethylene carbonate (FEC) has been commonly employed owing to its unique feature establishing a mechanically stable LiF-containing SEI, that can maintain the interfacial stability of Si-based anodes[11,22–24]. However, undesired defluorination of FEC by Lewis acidic $PF_5$ in $LiPF_6$-containing electrolytes, resulting in the generation of corrosive HF[25] and gaseous species such as $CO_2$[26,27], causes severe deterioration of storage performance of LIBs at high-temperature conditions. The use of FEC-containing electrolytes may require combination with complementary additives to ensure the desired action of FEC in LIBs. In particular, 1,3-dioxol-2-one, also known as vinylene carbonate (VC), has been commonly applied to form the SEI on the anode[20,21,28–32]. However, VC-derived SEIs comprising rigid poly(VC) species cannot bear the volumetric stress raised by the lithiation of Si[32–34]. Further, the structurally dense VC-derived SEIs act as resistive interfacial layers that hinder the fast charging performance of batteries and cause Li plating on the anode, which creates safety concerns[35,36]. More critically, the molecular-level synthetic design of functional VC derivatives has been challenging due to the destruction of the cyclic 1,3-dioxol-2-one nucleus associated with its labile electrophilic carbon center.

Silicon-centered, phosphorus-centered, or boron-centered compounds undergo electrochemical oxidation at Ni-rich cathodes prior to electrolyte decomposition, and contribute to the creation of a stable cathode-electrolyte interface (CEI). Therefore, they have been adopted to mitigate the interfacial damages of Ni-rich cathodes during cycling[37–39]. Further, the amelioration of electrochemical reversibility of Ni-rich cathodes has been accomplished using scavengers with basic electron-donating moieties, such as phosphite, amine, amino silane, and silyl ether. This is because the scavengers capture HF, which leaches out transition metal cations from the cathode and leads to the compositional change and structural damage of the SEI/CEI, which should be stably maintained to ensure the cycling stability of the electrodes[40].

Herein, we demonstrate the design and synthesis of functional VC derivatives bearing $-OCF_3$ and trimethylsilyloxy ($-OTMS$) moieties (Fig. 1) and report their application in LIBs comprising a high-capacity Si-embedded anode and a $LiNi_{0.8}Co_{0.1}Mn_{0.1}O_2$ (NCM811) cathode. A molecularly optimized SEI structure resolves the traditional drawbacks associated with VC-derived SEIs, such as rigidity, which hampers their reversible deformation upon Si volume expansion/contraction. The dimethylvinylene carbonate (DMVC)-bearing $-OCF_3$ group can act as an effective radical precursor through one-electron reduction, allowing successive propagation steps, and the $-OTMS$ moiety can effectively scavenge detrimental HF, provoking the destruction of the SEI/CEI. Our study revealed that the combination of VC, DMVC-$OCF_3$, and DMVC-OTMS offers a stable and deformable SEI on the Si–C anodes and maintains the interfacial stability of NCM811 cathodes through HF scavenging. Further, we show that the structural regulation of SEI and the improved stability of CEI with the use of DMVC-$OCF_3$, DMVC-OTMS, and VC enable fast charging of NCM811/Si-C full cells, which is vital for use in EVs.

## Results

**Retrosynthetic design of DMVC-$OCF_3$ and DMVC-OTMS.** Our retrosynthetic design of the additives based on the DMVC scaffold centers around the use of the $-OCF_3$ group as a fluorine source to generate LiF and the utilization of the $-OTMS$ group as an HF scavenger. DMVC-OH as a synthetic platform was prepared in 72% isolated yield in three steps, namely, radical bromination, formate ester generation, and hydrolysis (Fig. 1a; see also the Supplementary Methods and Supplementary Fig. 1)[41]. The synthetic route involving the formation of the readily hydrolysable formate ester intermediate was selected owing to the higher yield than under direct hydrolysis conditions (Supplementary Table 2 and Supplementary Fig. 2). DMVC-$OCF_3$ was then prepared by the silver-mediated $O$-trifluoromethylation of DMVC-OH to circumvent the instability issue associated with the use of the nucleophilic $CF_3O^-$ reagent (Fig. 1a)[42,43]. DMVC-OTMS was prepared by the $O$-silylation of DMVC-OH using chlorotrimethylsilane (TMSCl) and imidazole (Supplementary Table 1). Compared with the $^1H$ NMR chemical shifts of DMVC-OH, the $^1H$ NMR peaks of DMVC-$OCF_3$ were observed in the more deshielded region due to the reduced electron density from the trifluoromethyl moiety (Supplementary Fig. 3). The $^1H$ NMR spectrum of DMVC-OTMS clearly indicated a strong singlet peak at 0.17 ppm assigned to the trimethylsilane group. The characteristic quartet signal of the $\underline{C}F_3$ moiety of DMVC-$OCF_3$ in the $^{13}C$ NMR spectrum was observed along with carbonyl, vinyl, methyl, and methylene carbons. In addition, DMVC-OTMS provided the characteristic $^{13}C$ peaks, including those for Si $(\underline{C}H_3)_3$. The SEI is constructed on the Si–C anode via reductive copolymerization of DMVC-$OCF_3$, DMVC-OTMS, and VC during lithiation (Fig. 1b, c). Conversion of $COF_2$ with nucleophilic substances generated by the reductive decomposition of DMVC-$OCF_3$ may furnish the corresponding carbon dioxide or organic carbonate derivatives[44–46] (Supplementary Fig. 4). A possible mechanism for the improvement of the interfacial stability of Si–C anodes and the $LiNi_{0.8}Co_{0.1}Mn_{0.1}O_2$ (NCM811) cathodes by DMVC-$OCF_3$, DMVC-OTMS, and VC is depicted in Fig. 2.

**Confirmation of the copolymerization of VC derivatives.** The lowest unoccupied molecular orbital (LUMO) energy levels of DMVC-$OCF_3$ and DMVC-OTMS were lower than those of EC, VC, and fluoroethylene carbonate (FEC) (Fig. 3a and Supplementary Fig. 5), implying that DMVC-$OCF_3$ and DMVC-OTMS have a greater tendency for reduction at the anode than EC, VC, and FEC. Experimentally, the $dQ/dV$ graphs of the Li/Si–C half-cell confirmed that DMVC-$OCF_3$ and DMVC-OTMS had a higher reduction voltage than EC and VC, indicating that DMVC-$OCF_3$ and DMVC-OTMS modulated the interface

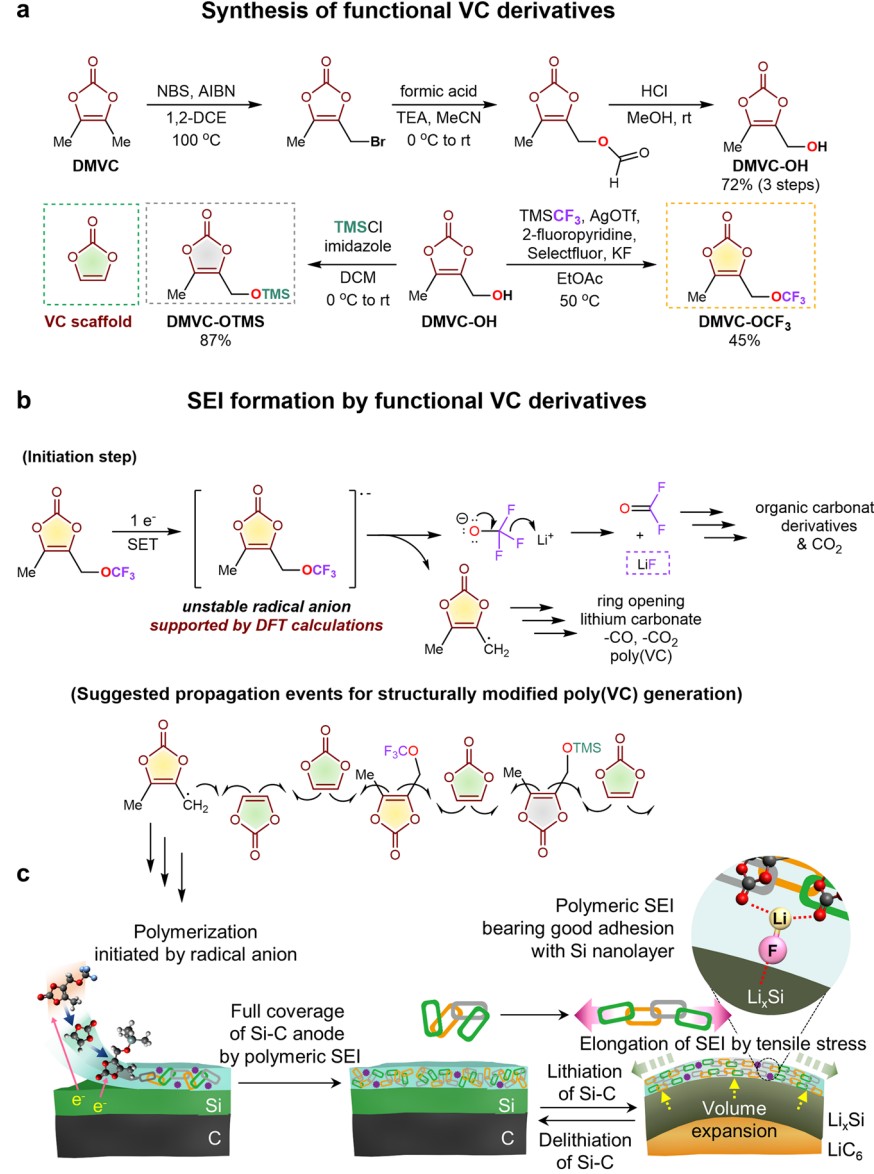

**Fig. 1 Synthesis of functional VC derivatives and transformation of additives to form SEI on the Si–C anode. a** Synthesis of DMVC-OTMS and DMVC-OCF₃. **b** Electrochemical transformations of DMVC derivatives. SET single electron transfer, NBS *N*-bromosuccinimide, AIBN azobisisobutyronitrile, 1,2-DCE 1,2-dichloroethane, TEA trimethylamine, MeCN acetonitrile; EtOAc ethyl acetate. **c** Design of a deformable and stable SEI using VC, DMVC-OCF₃ and DMVC-OTMS on the Si-C anode.

structure of the Si–C anode (Supplementary Fig. 6). Furthermore, we could predict that the decomposition of DMVC-OCF₃ into the DMVC radical and the OCF₃ anion by one-electron reduction occurred favorably (Fig. 3b) because the LUMO energy level of the OCF₃ radical was much lower than those of the DMVC radical and the decomposed DMVC-OCF₃ by C=C bond cleavage (Supplementary Fig. 7). The first lithiation of Li/Si-C half-cell also exhibits a reduction peak at 1.0 V vs. Li/Li⁺, which implies a one-electron reduction of DMVC-OCF₃ to produce the DMVC radical and OCF₃ anion (Supplementary Fig. 8a and c). The second peak (see Supplementary Fig. 8a) can be attributed to the reduction of the OCF₃ anion to form LiF because the LiF peak intensity was drastically increased after lithiation to 0.45 V (Supplementary Fig. 8a, b). Likewise, DMVC-OTMS showed a preference toward decomposition into DMVC radicals and OTMS anions rather than C=C bond cleavage (Supplementary Fig. 9). Among the species resulting from the decomposition of

DMVC-OCF₃, the OCF₃ anion could form LiF by interaction with Li⁺ ions. Note that in the presence of the Li⁺ ion, the activation energy and the heat of reaction decreased dramatically, facilitating LiF and OCF₂ formation (Fig. 3c). Because OCF₂ has a lower LUMO energy level than those of DMVC-OCF₃, the OCF₃ anion, and LiF (Supplementary Fig. 10), OCF₂ was more likely to accept an electron to form the OCF₂ anion (Fig. 3c). Promisingly, the C-centered DMVC radicals formed by the one-electron reductions of DMVC-OCF₃ and DMVC-OTMS underwent polymerization with the VC framework by attacking the olefinic carbon of the VC vinyl group, resulting in the formation of the SEI. Remarkably, our calculation results showed that the attack of the DMVC radical on VC was thermodynamically more favorable (Fig. 3d) than its reaction with DMVC-OCF₃ or DMVC-OTMS (Supplementary Fig. 11). The competitive decarboxylation route to release CO₂ was possible by the decomposition of the DMVC radical (Supplementary Figs. 12–14); however, the reaction of the

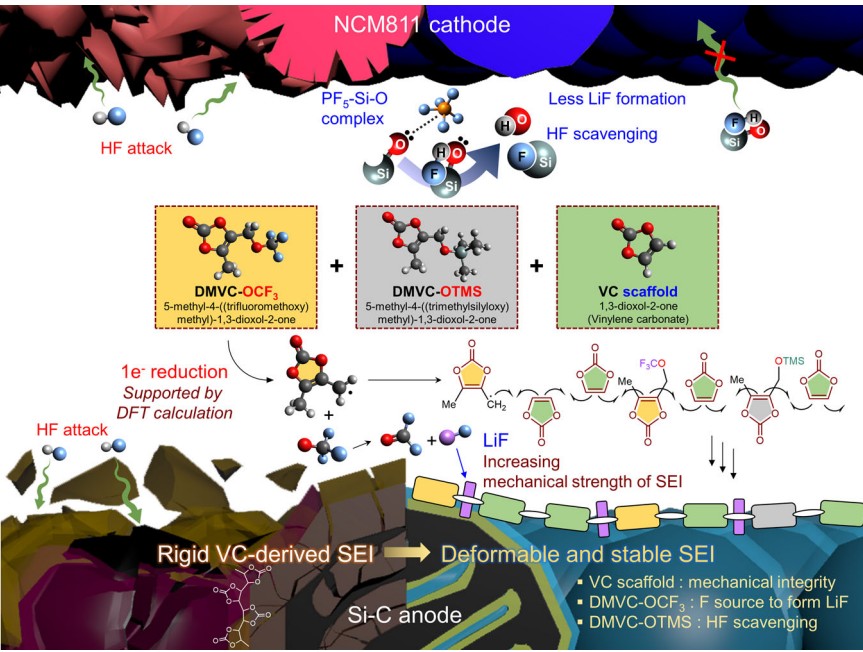

**Fig. 2 Unique features of DMVC-OCF₃, DMVC-OTMS, and VC for building stable interfacial layers.** Incorporation of DMVC-OCF₃ and DMVC-OTMS in the VC scaffold leads to the creation of a flexible and robust SEI on the Si–C anode. DMVC-OTMS scavenges HF and deactivates PF₅, resulting in compositional and structural stability of the interfacial layers on the electrodes. The Me (−CH₃) moiety bonded to the VC scaffold provides ion channels, providing space for Li-ion transport in the SEI.

DMVC radical with VC was the more dominant reaction because of its exothermicity and the low activation energy barrier compared to that for $CO_2$ generation (Supplementary Fig. 15). After the reaction of the DMVC radical with VC, the sequential reactions with DMVC-OCF₃, DMVC-OTMS, and VC were also exothermic while generating successive C-centered radicals (Fig. 3d). To probe the possibility of propagation reactions with various combinations of VC, DMVC-OCF₃, and DMVC-OTMS, we studied additional reactions with VC, DMVC-OCF₃, or DMVC-OTMS and found that most cases were thermodynamically favorable (Supplementary Fig. 16). Consequently, the reaction events with VC, DMVC-OCF₃, and DMVC-OTMS molecules were expected to propagate the creation of the polymeric SEI on the Si–C anode (Fig. 1b). To this end, we theoretically predicted that the DMVC radical and the OCF₃ anion from the reduction of DMVC-OCF₃ and DMVC-OTMS play a prominent role in the construction of the VC + DMVC-OCF₃ + DMVC-OTMS-derived SEI. A decrease in the C=C peak at 1650 $cm^{-1}$ and an increase in the C=O peak at 1775 $cm^{-1}$ via the copolymerization of VC, DMVC-OCF₃, and DMVC-OTMS were revealed through attenuated total reflectance Fourier transform infrared spectroscopy studies on the SEI (Supplementary Fig. 17). Furthermore, the C–F peak at 1180 $cm^{-1}$ appeared by the reductive decomposition of the OCF₃ anion.

**Electrochemical performance of NCM811/Si–C full cells.** The combination of VC, DMVC-OCF₃, and DMVC-OTMS enabled a high discharge capacity of 195.3 mAh g⁻¹ compared with additive-free electrolyte (179.0 mAh g⁻¹) during precycling (Supplementary Fig. 18). The initial Coulombic efficiency values of the full cells with VC + DMVC-OCF₃ + DMVC-OTMS were similar to those of the VC-containing and FEC-containing cells, indicating that VC + DMVC-OCF₃ + DMVC-OTMS forms suitable interfacial layers on both electrodes in full cells. The dQ/dV graphs of the full cells showed that VC + DMVC-OCF₃ + DMVC-OTMS contributed to SEI formation on the Si–C anode at 2.55 V, which is a lower potential than that for the VC

reduction potential of 2.90 V (Fig. 4a). Importantly, galvanostatic intermittent titration technique (GITT) experiments confirmed that the NCM811/Si-C full cell with VC + DMVC-OCF₃ + DMVC-OTMS exhibits reduced IR drop by less resistive interfacial layers compared with full cells with VC or FEC, allowing facile ion migration at high charge C-rates (Fig. 4b). Further, the impedance result of NCM811/Si–C full cell after 400 cycles revealed that VC + DMVC-OCF₃ + DMVC-OTMS made the SEI less resistive, leading to facile Li-ion transport (Supplementary Fig. 19). The cycle test of NCM811/Si–C full cells at 25 and 45 °C displayed distinct outcomes in their cycling performance (Fig. 4c and Supplementary Figs. 20 and 21). The NCM811/Si-C full cells without the additive showed severe capacity fading and low Coulombic efficiency over 400 cycles at 25 °C (Fig. 4c). The commonly used FEC for Si-embedded anodes had a better capacity retention (71.9%, Fig. 4d) than the VC (51.0%, Fig. 4e and Supplementary Fig. 22c). Although the use of DMVC-OCF₃, which can cross-couple to the VC framework via electrochemical copolymerization, improved the cycling stability of NCM811/Si–C full cells, the VC + DMVC-OCF₃ did not surpass the FEC ability. Notably, the VC + DMVC-OCF₃ + DMVC-OTMS attained stable cycling with an improved capacity retention (81.5%) after 400 cycles (Fig. 4c, f and Supplementary Fig. 22e). To determine the oxidation stability of DMVC-OCF₃ and DMVC-OTMS, the leakage current of Li/NCM811 half-cells was monitored at a constant charging voltage of 4.35 V vs. Li/Li⁺ for 3 h. Compared to FEC and VC, VC + DMVC-OCF₃ + DMVC-OTMS showed reduced leakage current, which indicates higher oxidation stability of the electrolyte. This result suggests that the presence of the C=C vinyl group of DMVC-OCF₃ and DMVC-OTMS does not negatively affect the performance of the NCM811 cathode at high potentials (Supplementary Fig. 23). Further, VC + DMVC-OCF₃ + DMVC-OTMS led to better cycling stability of full cells containing Si–C anodes with a higher Si content of 7 wt % than those of FEC and VC-added electrolytes (Supplementary Fig. 24). This result is enough to support the desirable effects of VC + DMVC-OCF₃ + DMVC-OTMS in LIBs compared with

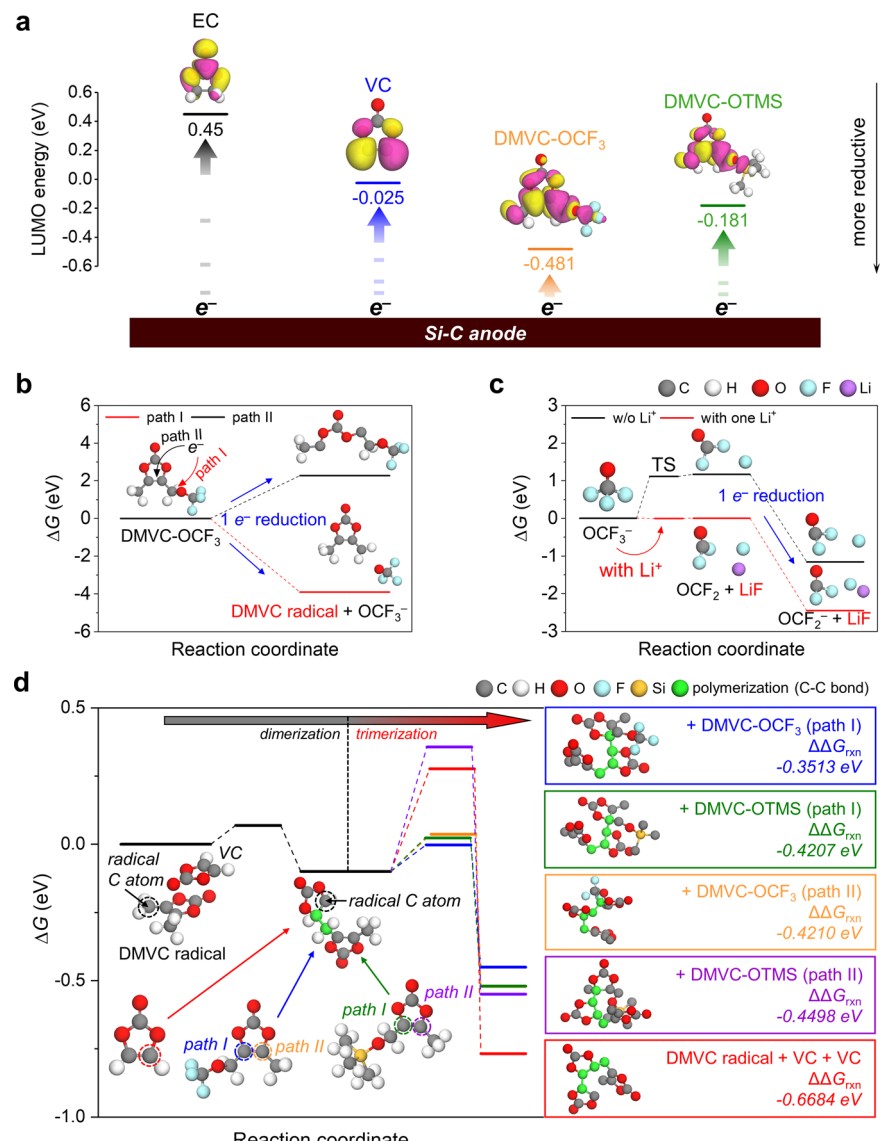

**Fig. 3 LUMO energy levels of solvent and additives and Gibbs free energies for the reduction of additives and the polymerization of additives. a** LUMO energy levels of EC, VC, DMVC-OCF$_3$, and DMVC-OTMS. Note that the isovalue of the orbital is 0.02 $e$/Å$^3$. **b–d** Reaction paths for the decomposition of DMVC-OCF$_3$ by one-electron reduction (**b**), decomposition of the OCF$_3$ anion (**c**), and polymerization by the DMVC radical with VC, DMVC-OCF$_3$, and DMVC-OTMS (**d**). Note that the relative Gibbs free energies ($\Delta G$) are calculated at 1 atm and 298 K. The $\Delta\Delta G_{rxn}$ in colored boxes indicates the difference in $\Delta G$s between the product and reactant of trimerization, which represents the heat of reaction in the trimerization reactions of the DMVC radical + VC dimer with VC, DMVC-OCF$_3$, or DMVC-OTMS. For the molecular structures in colored boxes, the hydrogen atoms are omitted for clarity.

previously reported results (Supplementary Table 3). In addition, NCM811/Si–C full cells with VC + DMVC-OCF$_3$ + DMVC-OTMS showed stable cyclability during 1000 cycles with 80% depth of discharge (Supplementary Fig. 25) and the improved capacity retention after 200 cycles at a C/5 rate (Supplementary Fig. 26). The NCM622/Si–C full cells and even NCM622/graphite full cells showed a better cycle performance with VC + DMVC-OCF$_3$ + DMVC-OTMS than the cells with FEC or VC, which demonstrates the broad applicability of the developed materials to other electrode systems (Supplementary Fig. 27). The proposed electrolyte system underwent undesired decomposition at the Li metal in half-cell configuration because of the stronger adsorption and high reactivity of DMVC-OCF$_3$ and DMVC-OTMS toward the Li metal (Supplementary Figs. 28–33). Therefore, high-quality SEI by VC + DMVC-OCF$_3$ + DMVC-OTMS was not formed on the Si–C anode in Li/Si–C half-cell, and the cathode-electrolyte interface was not maintained stably in the Li/NCM811 half-cell

because of parasitic reactions between the DMVC-OCF$_3$ and Li metal (Supplementary Figs. 34–37). The open circuit voltage (OCV) of the fully charged NCM811/Si–C full cell with FEC decreased considerably compared with that in case of VC and VC + DMVC-OCF$_3$ + DMVC-OTMS, and its capacity retention was significantly reduced to 60.7% after storing for 30 days at 60 °C (Supplementary Fig. 38). This result suggests that FEC-derived SEI and residual FEC, which is not consumed before storage experiment, are thermally unstable and do not restrain the self-discharge of a full cell. This is because of the undesirable defluorination of FEC, which produces HF and acid compounds that promote transition metal ion dissolution from the cathode[25]. The dissolved transition metal ions are then deposited on the anode surface by taking the electrons from the charged anode, and the electron loss of anode inevitably causes a reduction in the OCV and the capacity of charged full cells during storage at elevated temperatures.

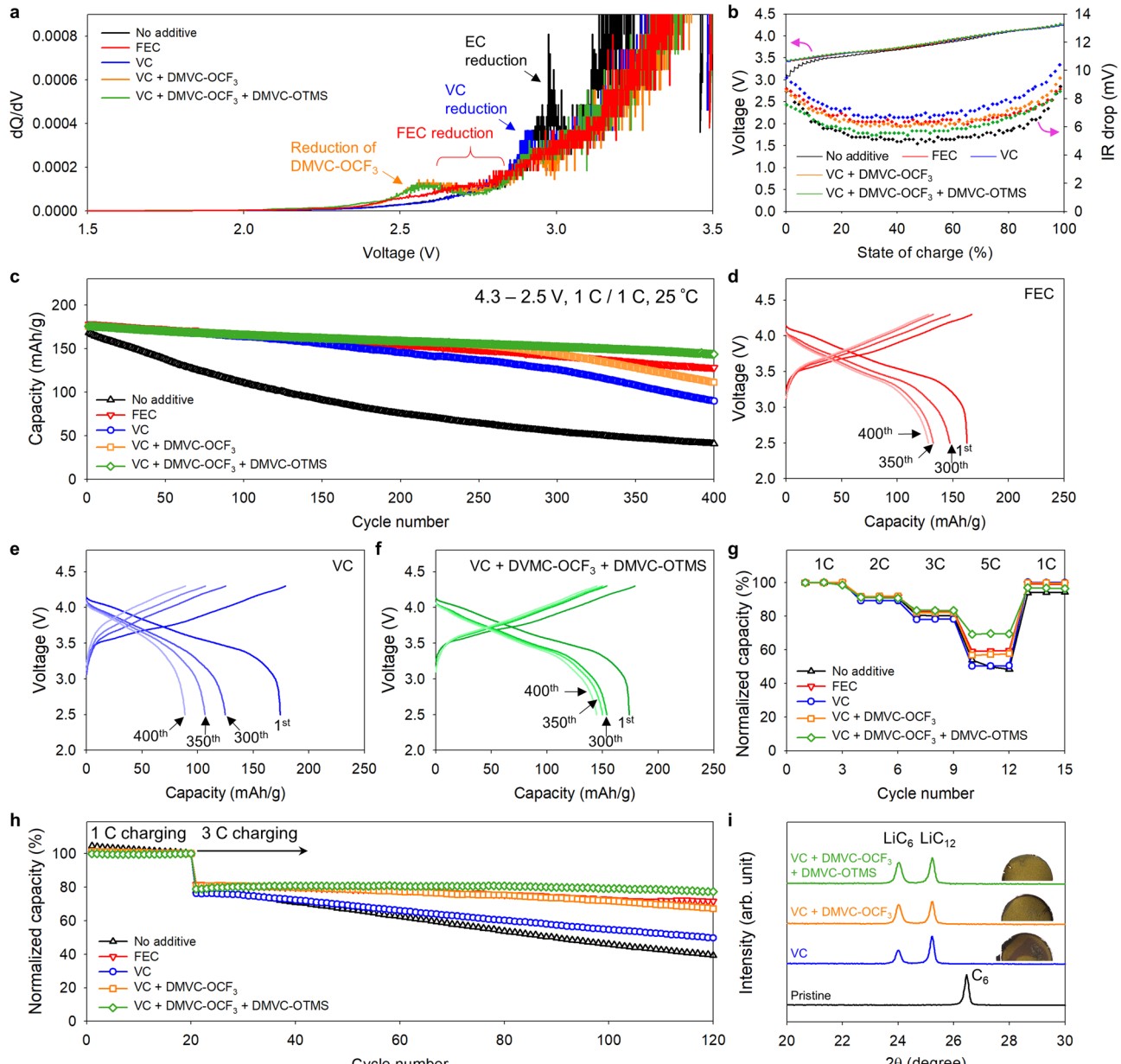

**Fig. 4 Electrochemical performance of synthesized functional additives and fast charging capability. a** d*Q*/d*V* graph of NCM811/Si–C full cells. (No additive: 1.15 M LiPF$_6$ in EC/EMC (3/7, v/v)) **b** Charge GITT profiles and IR drop of NCM811/Si–C full cells. **c** Cycle performance of NCM811/Si–C full cells at 1 C and 25 °C. **d–f** Voltage profiles of NCM811/Si–C full cells at a 1 C rate and 25 °C with FEC containing electrolyte (**d**), VC containing electrolyte (**e**), and VC + DMVC-OCF$_3$ + DMVC-OTMS containing electrolyte (**f**) at the 1st, 300th, 350th, and 400th cycles. **g** Charge rate capability of NCM811/Si–C full cells at a 1 C discharge rate. **h** Fast charging (1 C and 3 C) cycle performance of NCM811/Si–C full cells at a 1 C discharge rate at 25 °C. **i** XRD patterns and photographs of Si–C anodes charged (lithiated) at a 5 C rate.

**Enhanced fast charging capability.** To explore the suitability of the VC + DMVC-OCF$_3$ + DMVC-OTMS-derived SEI for facilitating Li-ion transport, we evaluated the cycling performance of NCM811/Si–C full cells at high charging rates (Fig. 4g). The VC + DMVC-OCF$_3$ + DMVC-OTMS resulted in superior discharge capacity at 5 C compared to that of cells containing VC alone. The X-ray diffraction (XRD) patterns of the Si-C anodes charged in the VC + DMVC-OCF$_3$ + DMVC-OTMS-containing electrolyte at 5 C exhibited a more pronounced peak of LiC$_6$ and a uniform gold color (Fig. 4i). This is attributed to the more ionically conductive SEI compared to the VC-promoted SEI, which showed a severely localized gold-colored Si–C anode with Li plating. Notably, the fast charging capability displayed at 3 C was

improved dramatically with the use of VC + DMVC-OCF$_3$ + DMVC-OTMS, and its capacity fading (1.9%) was negligible compared to that of VC (34.7%) (Fig. 4h). This result confirms that the synergistic combination of VC, DMVC-OCF$_3$, and DMVC-OTMS not only tolerates the volumetric stress of the Si–C anode, but also yields highly ion-conductive interfacial layers on both electrodes of the full cells.

**Conservation of the mechanical properties of SEI.** Comparative transmission electron microscopy (TEM) studies of the Si–C anodes with VC after precycling revealed that the Si nanolayer of the Si–C anode undergoes irreversible expansion (Fig. 5a, b).

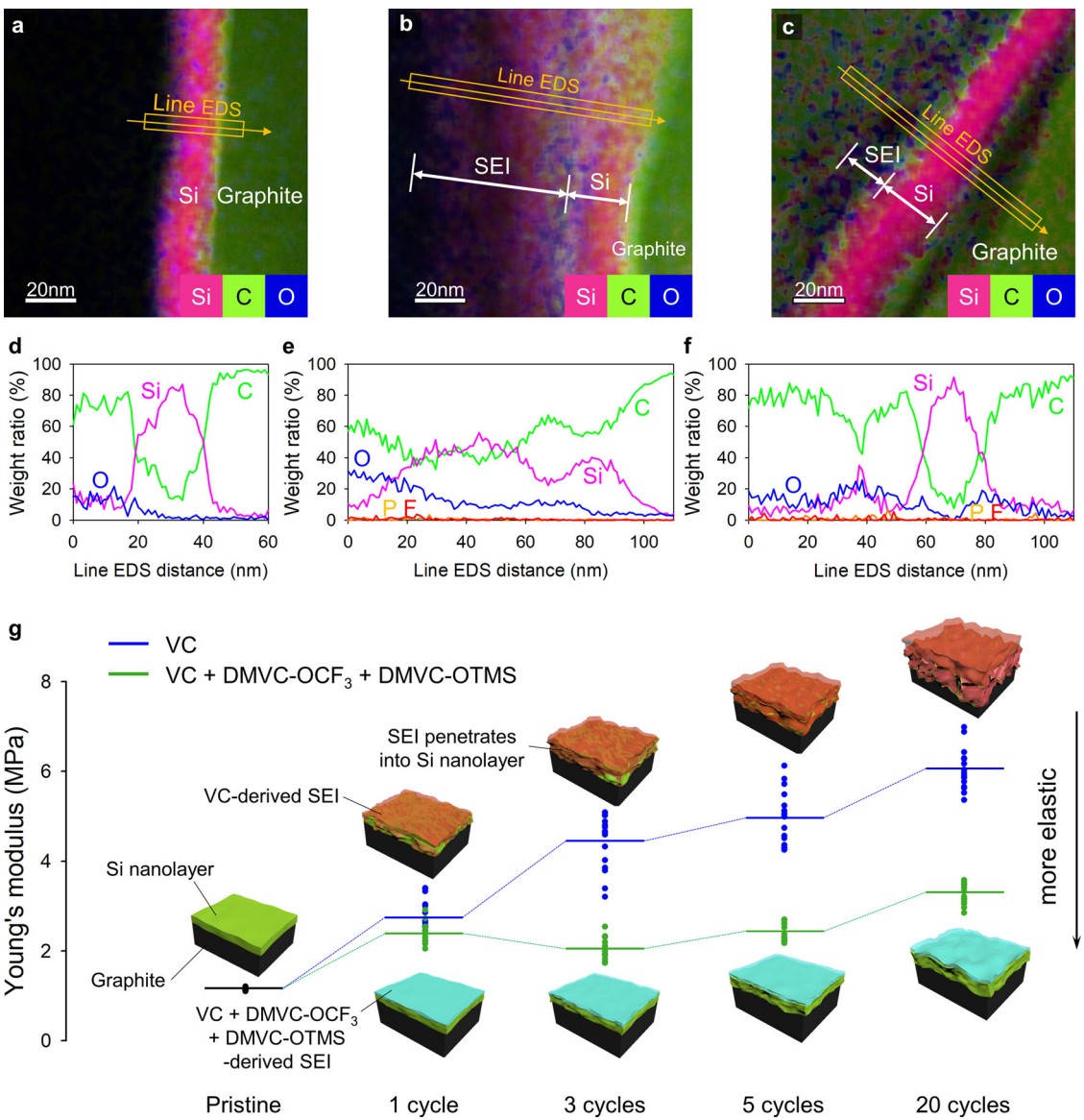

**Fig. 5 TEM characterization of the Si nanolayers of Si–C anodes after precycling of NCM811/Si–C full cells and Young's modulus of the Si nanolayers of Si–C anodes during cycling. a–c** TEM images and EDS mapping results (pink: silicon, green: carbon, and blue: oxygen) of a Si nanolayer of pristine Si–C (**a**), Si–C precycled in VC-added electrolyte (**b**), and VC + DMVC-OCF$_3$ + DMVC-OTMS-added electrolyte (**c**). **d–f** Line EDS-weighted profile of pristine Si–C (**d**), Si–C precycled with VC-containing electrolyte (**e**), and VC + DMVC-OCF$_3$ + DMVC-OTMS containing electrolyte (**f**). **g** Tendency of the Young's modulus of the Si nanolayer of the Si–C anode during cycling of NCM811/Si–C full cells.

Furthermore, the electrolyte decomposition byproducts containing carbon and oxygen permeated into the Si nanolayer of Si-C anodes with VC (Fig. 5d, e). By using VC, the Si nanolayer underwent an irreversible volumetric expansion and did not return to its original morphology after delithiation (Fig. 5b). In sharp contrast, the combined formulation of VC, DMVC-OCF$_3$, and DMVC-OTMS resulted in a Si nanolayer with well-maintained morphology after precycling (Fig. 5c). The line energy dispersive spectroscopy (EDS) spectra showed reduced penetration of the electrolyte decomposition byproducts into the Si nanolayer (Fig. 5f), because the VC + DMVC-OCF$_3$ + DMVC-OTMS-derived SEI maintained a stable structure with appropriate coverage to hinder severe damage to the anode surface.

To elucidate the roles of the SEI in the morphological stability of the Si nanolayer of the Si–C anode, nanoindentation by atomic force microscopy was performed[47–49] (Supplementary Fig. 39). The slope of the force curves from Si–C anodes cycled with VC

showed a continued increase during cycling (Supplementary Fig. 40a, b). The Young's modulus of the Si–C anode[50] (Supplementary Fig. 39b), was 1.15 MPa before cycling and increased to 6.0 MPa after 20 cycles with VC (Fig. 5g). On contrary, the Young's modulus of the Si–C anode cycled with VC + DMVC-OCF$_3$ + DMVC-OTMS was significantly lower than that for the anode cycled with VC alone. A lower Young's modulus indicates higher elasticity[51]; thus, the Si–C anode cycled with VC + DMVC-OCF$_3$ + DMVC-OTMS retains a more elastic SEI than the Si–C anode cycled with VC, which experienced penetration by the electrolyte decomposition byproducts. This elastic SEI is beneficial for enduring the volumetric stress; thereby, mechanical fracturing and the electrical isolation of Si are effectively mitigated.

The chemical structure of the VC + DMVC-OCF$_3$ + DMVC-OTMS-derived SEI was revealed via X-ray photoelectron spectroscopy (XPS) measurements. The peak intensity attributed to C–O,

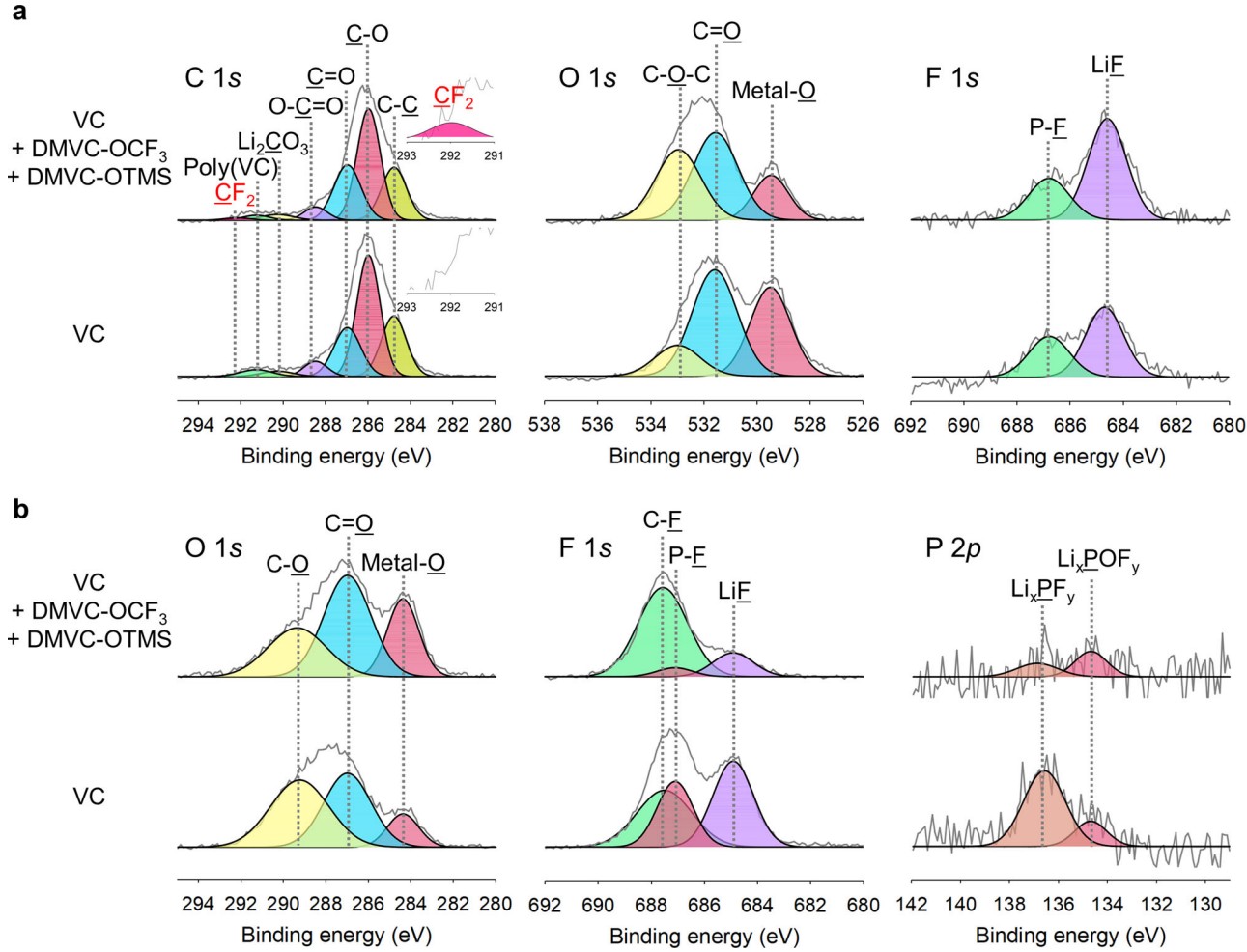

**Fig. 6 XPS spectra of Si–C anodes and NCM811 cathodes after precycling. a** C 1s, O 1s, and F 1s spectra for Si–C anodes after precycling of NCM811/Si–C full cells with VC + DMVC-OCF₃ + DMVC-OTMS and VC. **b**, O 1s, F 1s, and P 2p spectra for NCM811 cathodes obtained from NCM811/Si-C full cells after precycling in VC + DMVC-OCF₃ + DMVC-OTMS and VC.

C=O, and C–C species for the VC + DMVC-OCF₃ + DMVC-OTMS-derived SEI was similar to the VC-derived SEI because of the similarity of their frameworks (C 1s XPS in Fig. 6a). The CF₂ peak, which might be formed by the reduction of the OCF₃ anion, appeared at 292 eV in the case of VC + DMVC-OCF₃ + DMVC-OTMS (Supplementary Table 4). A noticeable feature of the Si-C anode with VC + DMVC-OCF₃ + DMVC-OTMS is that the peak intensity associated with the C=O and metal-O decreased drastically (O 1s XPS in Fig. 6a and Supplementary Table 5). This result implies that DMVC-OCF₃ and DMVC-OTMS modify the structure of the VC-derived SEI. Notably, the LiF peak intensity substantially increased at the SEI on the Si–C anodes precycled in VC + DMVC-OCF₃ + DMVC-OTMS (Supplementary Table 6). This is attributable to the decomposition of OCF₃⁻ generated by the reduction of DMVC-OCF₃. The metal-O peak at 529.5 eV increased noticeably (Supplementary Table 7), likely because a thinner CEI is formed on the cathode surface with VC + DMVC-OCF₃ + DMVC-OTMS. The peaks corresponding to LiF and the P-F moiety in the CEI were of remarkably lower intensity in VC + DMVC-OCF₃ + DMVC-OTMS than those in VC (Fig. 6b and Supplementary Table 8). The XPS analysis of the cathodes after precycling clearly indicates that the LiPF₆ decomposition and the LiF formation at the cathode are

suppressed by VC + DMVC-OCF₃ + DMVC-OTMS (Fig. 6b and Supplementary Table 9).

The Si–C anode cycled with the VC had severely cracked particles, indicating the loss of their electrical connection (Fig. 7a, b). The exposure of the active surface of the Si–C anode particles leads to continuous electrolyte decomposition, causing thickening of the SEI to block Li-ion transfer and electron movement between the Si-C anode particles. The feature on the Si–C anode with VC + DMVC-OCF₃ + DMVC-OTMS was strikingly different. The morphology of the Si–C anode particles was intact without any clear signs of mechanical fracture (Fig. 7c). This finding reveals that VC + DMVC-OCF₃ + DMVC-OTMS forms a multifunctional SEI that accommodates the strain raised by repeated lithiation and delithiation of the Si–C anode, and effactually protects the Si–C anode against HF attack and transition metal deposition. The Si–C anode with VC showed an enormous volume expansion of approximately 176% after 400 cycles (Fig. 7d–f). Importantly, VC + DMVC-OCF₃ + DMVC-OTMS effectively alleviated the increase in the thickness of Si–C anodes compared to that with VC alone (Fig. 7f). The EDS mapping images in TEM of Si–C anodes after 400 cycles demonstrated the severe volume expansion of Si with VC (Fig. 7h and Supplementary Fig. 42e). The C and O EDS

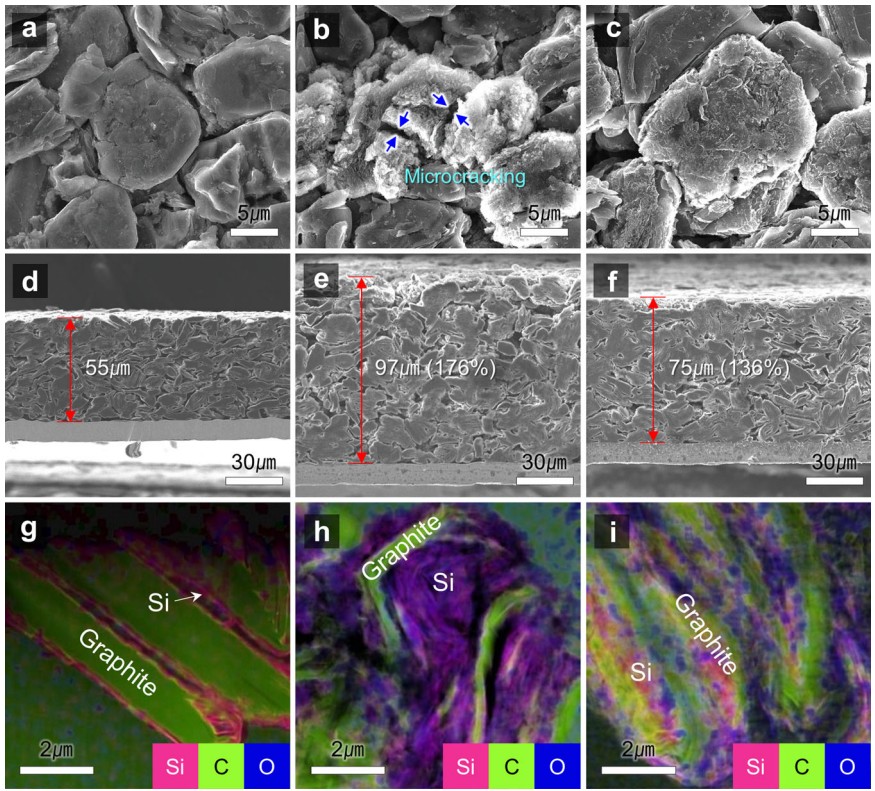

**Fig. 7 SEM and TEM characterization of Si–C anodes after 400 cycles of NCM811/Si–C full cells at 25 °C. a–f** Surface morphologies of a pristine Si–C anode (**a**) and Si–C anodes obtained from NCM811/Si–C full cells cycled during 400 cycles at 25 °C with VC (**b**) or VC + DMVC-OCF₃ + DMVC-OTMS (**c**), cross-sectional views of the pristine Si–C anode (**d**) and Si–C anodes from NCM811/Si–C full cells cycled during 400 cycles at 25 °C with VC (**e**) or VC + DMVC-OCF₃ + DMVC-OTMS (**f**). **g–i** EDS mapping in TEM of the pristine Si–C anode (**g**) and Si–C anodes after 400 cycles with VC (**h**) or VC + DMVC-OCF₃ + DMVC-OTMS (**i**).

mapping images of Si–C anodes precycled with VC showed that the electrolyte decomposition byproducts permeate into the Si nanolayer (Supplementary Fig. 41c, d). In contrast, the Si nanolayer of Si–C anodes with VC + DMVC-OCF₃ + DMVC-OTMS stably maintained its original layered structure without irreparable damage (Fig. 7g, i and Supplementary Fig. 42f). Additionally, the SEI fabricated using VC + DMVC-OCF₃ + DMVC-OTMS was thinner than the VC-derived SEI (Supplementary Fig. 41a, e).

Corrosive HF causes the undesired elution of transition metal cations from the cathode and the irreversible deposition of leached transition metal cations on the anode surface[52,53]. Moreover, HF severely damages the CEI and SEI structures that must be maintained throughout the charge–discharge cycles to protect the anodes and cathodes[54]. To elucidate the vital role of DMVC-OTMS in HF scavenging, 1 wt% water was introduced to the additive-free and DMVC-OTMS-containing electrolytes, and the solutions were kept in storage for 1 day at 25 °C. The ¹⁹F NMR spectra of the additive-free electrolyte with 1% water shows peaks near −193.9 and −85.1 ppm that could be assigned to HF and $PO_2F_2^-$, respectively (Supplementary Fig. 43a). Additionally, $PO_3F^{2-}$, which was formed by the subsequent conversion of $PO_2F_2^-$, was detected in the ³¹P NMR spectrum (Supplementary Fig. 43c). As expected, the DMVC-OTMS-containing electrolyte with 1% water did not show the characteristic resonance of HF at −193.9 ppm (Supplementary Fig. 43b) and those for $PO_2F_2^-$ and $PO_3F^{2-}$ (Supplementary Fig. 43d). This result provides strong evidence that DMVC-OTMS effectively scavenges HF and prevents the

sequential hydrolysis of $LiPF_6$ to $HPO_2F_2$ and $H_2PO_3F$ (Supplementary Fig. 43e).

The impact of additives on the extent of transition metal deposition on the cycled Si–C anodes was examined using inductively coupled plasma–optical emission spectroscopy (Supplementary Table 10). After 400 cycles, the Si–C anode cycled with VC + DMVC-OCF₃ + DMVC-OTMS showed a further reduction in the amount of Ni deposited on the surface (37.7 ppm), which was lower than the 55.2 ppm of Ni deposited on the Si–C anode cycled with VC + DMVC-OCF₃, thus revealing the suppression of the transition metal dissolution effect of DMVC-OTMS via HF scavenging.

In conclusion, we demonstrated that the creation of a stable and spatially deformable SEI on a high-capacity Si–C anode could tolerate the inevitable volume changes induced by the lithiation of Si and could enable a long lifespan and fast chargeability of high-energy-density lithium-ion batteries. DMVC-OCF₃ prepared by silver-mediated O-trifluoromethylation of DMVC-OH initiated the facile construction of the flexible and robust SEI on the Si–C anode while producing LiF as a mechanical enhancer of the SEI. Notably, HF, which severely damages the CEI and SEI layers, was effectively scavenged by the OTMS group in DMVC-OTMS; thereby, the structural integrity of the CEI and SEI layers was preserved. This work presents a breakthrough in the development of electrolyte additives for high-energy-density Li-ion batteries. We expect that our systematic approach for rational molecular design and DFT-aided mechanism development offers a promising way to discover next-generation additives.

## Methods

**Synthesis of 5-methyl-4-((trifluoromethoxy)methyl)-1,3-dioxol-2-one (DMVC-OCF₃).** 2-Fluoropyridine (388 mg, 4.0 mmol, 2.0 equiv) and TMSCF₃ (569 mg, 4.0 mmol, 2.0 equiv) were added to a mixture of AgOTf (1.03 g, 4.0 mmol, 2.0 equiv), Selectfluor (1.06 g, 3.0 mmol, 1.5 equiv), KF (350 mg, 6.0 mmol, 3.0 equiv), and DMVC-OH (260 mg, 2 mmol, 1.0 equiv) in ethyl acetate (10 mL) under an inert Ar atmosphere. The reaction mixture was stirred at 50 °C for 12 h. The reaction mixture was filtered, concentrated, and purified by flash column chromatography over silica gel (5:l, ethyl acetate/n-hexane) to afford a yellow oil-like title compound (178.3 mg, 0.9 mmol, 45%). $^1$H NMR (400 MHz, CDCl₃) $\delta$ 4.73 (s, 2H), 2.18 (s, 3H); $^{13}$C NMR (101 MHz, CDCl₃) $\delta$ 151.6, 140.9, 131.8, 121.4 (q, $J =$ 257.7 Hz), 56.8 (q, $J =$ 4.1 Hz), 9.3; $^{19}$F NMR (377 MHz, CDCl₃) $\delta$ −60.85 (s). HRMS (ESI+) $m/z$ calculated for $C_6H_6F_3O_4$ ([M+H]$^+$) 199.0213, found 199.0214.

**Synthesis of 5-methyl-4-((trimethylsilyloxy)methyl)-1,3-dioxol-2-one (DMVC-OTMS).** TMSCl (261 mg, 2.4 mmol, 1.2 equiv) was added to a mixture of imidazole (340 mg, 5 mmol, 2.5 equiv) and DMVC-OH (260 mg, 2.0 mmol, 1.0 equiv) in argon-purged dichloromethane (10 mL) at 0 °C. The reaction mixture was stirred at room temperature for 12 h and was diluted with brine. The product was extracted with dichloromethane. The combined organic layers were dried over Na₂SO₄ and concentrated in vacuo to afford the title compound as an orange liquid (352 mg, 1.74 mmol, 87%). $^1$H NMR (400 MHz, CDCl₃) $\delta$ 4.37 (s, 2H), 2.12 (s, 3H), 0.17 (s, 9H); $^{13}$C NMR (101 MHz, CDCl₃) $\delta$ 153.2, 137.6, 137.4, 54.0, 9.9, 0.0; HRMS (ESI+) $m/z$ calculated for $C_8H_{15}O_4Si$ ([M+H]$^+$) 203.0734, found 203.0731.

**Electrolyte and electrode preparation.** The baseline electrolyte was 1.15 M LiPF₆ in ethylene carbonate (EC) and ethyl methyl carbonate (EMC) (3:7 vol%). Then, 5 wt% fluoroethylene carbonate (FEC), 1.5 wt% VC or 0.5 wt% vinylene carbonate (VC) + 0.5 wt% DMVC-OCF₃ + 0.5 wt% DMVC-OTMS were added into the baseline electrolyte for evaluation. To minimize the water content, CaH₂ was incorporated to the electrolytes and stirred for 30 min, followed by filtration. VC, FEC, and electrolyte solvents were purchased from Soulbrain Co., Ltd. (South Korea). The Ni-rich cathode was fabricated by spreading a slurry composed of 92.5 wt% LiNi₀.₈Co₀.₁Mn₀.₁O₂ (Single crystalline NCM811, SMLAB (South Korea)), 3.5 wt% conducting agent (2 wt% carbon black (Super C65, Imerys Graphite & Carbon) + 1.5 wt% graphite (SFG6L, Imerys Graphite & Carbon)), and 4 wt% binding material (poly(vinylidene fluoride), Solef6020, Solvay) in 1-methyl-2-pyrrolidinone (Sigma-Aldrich) on Al foil (15 µm). The cathode prepared drying the slurry at 120 °C for 30 min was pressed by a roll press machine[15]. The areal capacity and loading level of the cathode with the thickness of 44 µm were 2.7 mA h cm$^{-2}$ and 13.5 mg cm$^{-2}$, respectively. The Si–C anode was composed of 37.4 wt% Si nanolayer-embedded graphite (SNG with 7 wt% Si, SJ Advanced Materials), 58.6 wt% graphite (LA1, Shanshan (China)), 1 wt% carbon black (Super C65, Imerys Graphite & Carbon), and 3 wt% binding material (2 wt% styrene-butadiene rubber (BM-400B, Zeon) + 1 wt% carboxymethyl cellulose (MAC350H, Nippon Paper Group)) in distilled water and coated onto a Cu foil (10 µm). The SNG was fabricated using a chemical vapor deposition (CVD) process according to literature[55]. The specific capacity and content of Si of the Si–C anode based on the SNG/graphite composite were 435.7 mAh g$^{-1}$ and 3 wt%, respectively. The anode was also pressed by a roll press machine. The anode with the thickness of 55 µm had a areal capacity of 3.2 mA h cm$^{-2}$ and mass loading of 7.5 mg cm$^{-2}$. The EDS mapping spectra of the pristine Si–C anodes showed the presence of a Si nanolayer with a thickness of approximately 20 nm coated on the graphite to form the Si–C anode (Fig. 5a, d). To eliminate water, the electrodes were dried at 110 °C for 10 h under vacuum before use in cell fabrication. A 20 µm thick and 38% porosity polyethylene membrane (SK Innovation Co., Ltd.) was adopted as a separator.

**Electrochemical measurements.** Two thousand and thirty-two round-type full cells were fabricated in an argon-filled glove box, and an N/P ratio of 1.3 was determined using Eq. 1.

$$\frac{(\text{Discharge capacity of anode}) \times (\text{Mass of anode})}{(\text{Discharge capacity of cathode}) \times (\text{Mass of cathode}) - (\text{Irreversible capacity of anode}) \times (\text{Mass of anode})} =$$

$$\frac{(420.5\,\text{mA h/g}) \times (7.5\,\text{mg/cm}^2)}{(201.4\,\text{mAh/g}) \times \left(13.5\,\text{mg/cm}^2\right) - (40.9\,\text{mAh/g}) \times (7.5\,\text{mg/cm}^2)} = 1.3 \tag{1}$$

The amount of electrolyte per capacity was 27.7 mg mAh$^{-1}$. The full cells composed of the Ni-rich cathode and Si–C anode were galvanostatically cycled in a voltage range between 4.3 V and 2.5 V at 25 °C (WBCS3000, WonATech). Precycling for the formation of the SEI and CEI was performed at C/5 once. The cells were charged up to 4.3 V at C/5 followed by a constant voltage (CV) phase with a C/20 current cutoff; then, they were discharged to 2.5 V at 25 °C. Standard cycles with C/5 and C/2 for one time each were performed between 4.3 and 2.5 V at 25 °C for subsequent cycling. A C/20 current cutoff was applied to finish the CV condition of the charge process. The GITT experiment was performed after two standard cycles (C/5 rate and C/2 rate once each). The cells were charged up to 4.3 V at C/5 for 5 min and then were left to rest for 30 min to attain equilibrium voltage. A cycle test was performed without a CV condition at 1 C at both 25 and 45 °C (1 C = 2.7 mA cm$^{-2}$). The charge rate capability evaluation of full cells was conducted at a fixed discharge rate of 1 C and various charge C-rates (1, 2, 3, and 5 C).

**Characterization.** The cycled electrodes for analysis of the surface chemistry, mechanical properties, and morphology were obtained from the full cells disassembled in a glove box. The residual electrolyte from the retrieved electrodes was removed using dimethyl carbonate (DMC) solvent. The SEI structure on the Si–C anode was identified via XPS (Scientific K-Alpha system, Thermo Scientific) with Al Kα radiation ($hv =$ 1486.6 eV). All XPS spectra were energy calibrated by the hydrocarbon peak at 284.8 eV. To verify the scavenging effect of DMVC-OTMS on HF, $^{19}$F nuclear magnetic resonance spectroscopy (400 MHz FT-NMR (Bruker), AVANCE III HD) was performed. DI water (1 wt%) was incorporated to the baseline electrolyte and to the electrolyte with 1 wt% DMVC-OTMS followed by storage at 25 °C for 24 h. The stored electrolytes were analyzed by NMR using tetrahydrofuran-d₈ solvent. The mechanical analysis was examined by the AFM nanoindentation method (details are described in the Supplementary information, MultiMode V, Veeco). Morphological and structural changes of the anodes were confirmed using field emission scanning electron microscopy (FE-SEM, JSM-6700F, JEOL) with high-resolution transmission electron microscopy (HR-TEM, JEM-2100F, JEOL).

**Computational details: density functional theory (DFT) calculations.** In this study, we investigated the LUMO energy levels, reaction mechanisms, and adsorption energies using DFT calculations. All DFT calculations were carried out using the DMol[3] program[56,57] under an implicit environment by using the conductor-like screening model (COSMO) method[58] with a dielectric constant of 13.287 (3:7 mixture of EC (95.3)[59] and EMC (2.9)[59] at 25 °C through the mixing rule[60]). The detailed information is described in the Supplementary information.

## Data availability

The authors declare that the main data supporting the findings in this study are available within the article and its Supplementary information. Additional data are available from the corresponding authors upon reasonable request.

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

## Acknowledgements

This work was supported by the Korea Institute of Energy Technology Evaluation and Planning/IT R&D program of the Ministry of Trade, Industry & Energy (MOTIE/KETEP) under Project Number 20172410100140. This research was partly supported by the Technology Development Program to Solve Climate Changes of the National Research Foundation (NRF), funded by the Ministry of Science, ICT & Future Planning (2018M1A2A2063341).

## Author contributions

S.P., S.Y.J., T.K.L., and M.W.P. contributed equally to this work. S.P., S.Y.J., S.Y.H., and N.-S.C. proposed and designed the project. S.P. and M.W.P. performed electrochemical measurements and carried out characterizations. S.Y.J. and S.Y.H. synthesized the functional electrolyte additives (DMVC-OCF3 and DMVC-OTMS). T.K.L., H.Y.L., and S.K.K. performed the density functional theory calculations. J.S. and J.C. synthesized and provided the electrode materials. S.P., S.Y.J., T.K.L., S.K.K., S.Y.H., and N.-S.C. wrote the manuscript. All authors discussed the results and commented on the manuscript.

## Competing interests

The authors declare no competing interests.
