## [Peer Review File · Nature Communications]

Reviewer #1 (Remarks to the Author):

The authors report the synthesis of two new additives based on a VC backbone, one functionalized with a trifluoromethoxy group and one with a trimethylsilyloxy group. They further test these additives in a conventional baseline electrolyte containing EC, EMC, and LiPF₆ and compare the performance of Si/C-graphite/NMC811 full cells employing to that obtained with only VC or FEC as additives. The proposed electrolyte significantly improves the cycling stability and rate performance compared to the other electrolytes. Furthermore, the authors conducted a mechanistic study, making use of extensive DFT calculations and various experimental techniques. The results suggest the formation of a more flexible SEI compared to VC and to the ability of the electrolyte to scavenge HF. The results are novel and of high interest to the community. The manuscript could be considered for publication after the following comments have been addressed:

- 1) The introduction should give a brief overview of the recent developments of additives for Si-based anodes and Ni-rich NMC cathodes to put the work in the context of the efforts of the community. FEC should be mentioned already in the introduction as this is the most prominent additive for Si-based anodes. Additives that stabilize Ni-rich NMC cathodes should also be mentioned.
- 2) I was surprised to find out in the supplementary information that the employed anode contains only 3% Si. This fact should be clearly stated in the main part of the manuscript.
- 3) More details such as supplier, composition, etc. should be given about the employed Si-C material.
- 4) The cell formation protocol also has to be disclosed in full detail including number of cycles at each rate, voltage range, use of constant voltage step, temperature etc.
- 5) How effectively does the proposed electrolyte stabilize electrodes with a higher Si content? The authors should compare the cycling stability of Si-rich anodes, e.g. anodes only containing the Si-C active material, using the multi-additive electrolyte to that using the FEC electrolyte.
- 6) To understand whether the proposed additive combination is superior in stabilizing Si-based anodes compared to FEC, half-cell tests vs. Li metal should be conducted for the Si-C anode in the proposed electrolyte and in the FEC-based electrolytes.
- 7) Half-cell tests should also be conducted for the NMC811 cathode to understand the influence of the novel additives on the stabilization of the cathode compared to VC and FEC.
- 8) The electrolytes should also be compared at a low rate such as C/5.

Reviewer #2 (Remarks to the Author):

The performance of VC + DMVC-OCF₃ + DMVC+ OTMS is only slightly better than the cells with FEC or VC. As there is no statistically convincing data to support that this new electrolyte is really making a significant difference, I don't have the confidence to support the acceptance of this article.

An inherent challenge of this work is that the study has treated this battery as a black box. The overall performance is evaluated without paying any attention to the performance of the cathode or anode alone. However, the argument of the positive impacts of this electrolyte is based on postulations pertaining to individual electrodes. This becomes a tautological argument.

I hope the following suggestions are useful for the authors to improve their manuscript. Please provide the potential profiles of the cathode and anode at different stages of the cycling. Particularly, I hope to see the profiles for the failing cycles at the end of the cycling.

What about the cycling stability of both anode and cathode in half cells?

What is the specific capacity of the cathode and anode in half cells?

What is the N/P ratio?

The font of the words in Figure 1c looks strange. Please fix it.

Figure 3a, the LUMO energy level of EC is positive. What is the reference?

Figure 7e, why with VC + DMVC-OCF3 + DMVC-OTMS, the thickness of the Si-C anode is much bigger?

Reviewer #3 (Remarks to the Author):

In this work, two functional VC derivatives of DMVC-OCF3 and DMVC94-OTMS are synthesized and studied as the electrolyte additive for a Si/C-NMC811 cell. The results are meaningful, and might be suitable for publication after revising and clarifying the questions below:

1. Bromine in DMVC-Br is relatively active, which should be easily hydrolyzed. Why the synthesis of DMVC-OH do not use the direct hydrolysis of DMVC-Br, instead, a two-step synthesis route?
2. In Fig. 1b, the O=CF₂ product is a gas and highly reactive to all of the SEI, CEI, solvents, and electrode materials (Si and NMC811), it cannot be stably present in the cell. Please propose a reasonable final product.
3. Both DMVC-OCF₃ and DMVC94-OTMS contain >C=C< vinyl groups that are unstable at high potentials. How they affect the performance of NMC811 cathode in addition to HF-scavenge? The oxidation of the >C=C< vinyl compounds on the cathode side (at high potentials) should be studied separately.
4. In Supplementary Fig. 3a, there are two dQ/dV peaks for DMVC-OCF₃, what reduction does the other peak at higher potential correspond to? Besides, DMVC-OTMS chart is missed in Supplementary Fig. 3b.
5. In Fig. 4b, R_{sei} and R_{ct} are changed within the error ranges, especially the right two (they have almost the same R_{sei} and R_{ct}). It appears that the impedance results cannot explain so large differences in the capacity at 5C in Fig. 4d.
6. More details are needed in experimental: What potential/voltage ranges were used for the measurements of NMC811 (2.7 mA h cm⁻²) and anode (3.2 mA h cm⁻²) loadings? Is the Si-C a mixture or a Si-coated C composite? If it is a Si-C mixture, why there are no gaps between the Si and graphite boundaries in the TEM images (Figs. 5a-c); What is the Si content in Si-C mixture/composite?

Manuscript ID: NCOMMS-20-27266-T

Manuscript Type: Article

Title: Replacing conventional battery electrolyte additives with dioxolone derivatives for high-energy-density lithium-ion batteries

Dear Editor and Reviewers,

We would like to thank you for their critical and helpful comments that have helped us revise and improve our manuscript. Below, we present our point-by-point responses to all the questions, along with our modifications (highlighted in light blue) in the revised manuscript.

Reviewer #1 (Remarks to the Author):

The authors report the synthesis of two new additives based on a VC backbone, one functionalized with a trifluoromethoxy group and one with a trimethylsilyloxy group. They further test these additives in a conventional baseline electrolyte containing EC, EMC, and LiPF₆ and compare the performance of Si/C-graphite/NMC811 full cells employing to that obtained with only VC or FEC as additives. The proposed electrolyte significantly improves the cycling stability and rate performance compared to the other electrolytes. Furthermore, the authors conducted a mechanistic study, making use of extensive DFT calculations and various experimental techniques. The results suggest the formation of a more flexible SEI compared to VC and to the ability of the electrolyte to scavenge HF. The results are novel and of high interest to the community. The manuscript could be considered for publication after the following comments have been addressed:

1. The introduction should give a brief overview of the recent developments of additives for Si-based anodes and Ni-rich NMC cathodes to put the work in the context of the efforts of the community. FEC should be mentioned already in the introduction as this is the most prominent additive for Si-based anodes. Additives that stabilize Ni-rich NMC cathodes should also be mentioned.

Response. We thank the reviewer for the valuable comment on our manuscript. Recent literatures^{R1-R13} focused on electrolyte additives have been cited in the introduction section in the main text to provide an overview associated with the interfacial deterioration of Si-based anodes and Ni-rich layered oxide cathodes.

On page 3 (main text): So far, reductive compounds possessing fluorine-donating moiety or vinyl group¹⁸⁻²¹ have been exploited as SEI-forming additives for Si-based anodes. Fluoroethylene carbonate (FEC) has been commonly employed owing to its unique feature establishing a mechanically stable LiF-containing SEI that can maintain the interfacial stability of Si-based anodes^{11,22-25}. However, undesired defluorination of FEC by Lewis acidic PF₅ in LiPF₆-containing electrolytes, resulting in the generation of corrosive HF²⁶ and gaseous species such as CO₂^{27,28}, causes severe deterioration of storage performance of LIBs at high-temperature conditions. The use of FEC-containing electrolytes may require combination with complementary additives to ensure the desired action of FEC in LIBs.

On page 4 (main text): Silicon-, phosphorus-, or boron-centered compounds undergo electrochemical oxidation at Ni-rich cathodes prior to electrolyte decomposition and contribute to the creation of a stable cathode-electrolyte interface (CEI). Therefore, they have been adopted to mitigate the interfacial damages of Ni-rich cathodes during cycling³⁸⁻⁴⁰. Further, the amelioration of electrochemical reversibility of Ni-rich cathodes has been accomplished using scavengers with basic electron-donating moieties, such as phosphite, amine, amino silane, and silyl ether. This is because the scavengers capture HF, which leaches out transition metal cations from the cathode and leads to the compositional change and structural damage of the SEI/CEI, which should be stably maintained to ensure the cycling stability of the electrodes⁴¹.

References

- R1.** Jo, H. *et al.* Stabilizing the solid electrolyte interphase layer and cycling performance of silicon-graphite battery anode by using a binary additive of fluorinated carbonates. *J. Phys. Chem. C* **120**, 22466–22475 (2016).
- R2.** Nguyen, C. C. & Lucht, B. L. Improved cycling performance of Si nanoparticle anodes via incorporation of methylene ethylene carbonate. *Electrochem. Commun.* **66**, 71–74 (2016).
- R3.** Etacheri, V. *et al.* Effect of fluoroethylene carbonate (FEC) on the performance and surface chemistry of Si-nanowire Li-ion battery anodes. *Langmuir* **28**, 965–976 (2012).
- R4.** Choi, N.-S. *et al.* Effect of fluoroethylene carbonate additive on interfacial properties of silicon thin-film electrode. *J. Power Sources* **161**, 1254–1259 (2006).
- R5.** Xu, C. *et al.* Improved performance of the silicon anode for li-ion batteries: Understanding the surface modification mechanism of fluoroethylene carbonate as an

- effective electrolyte additive. *Chem. Mater.* **27**, 2591–2599 (2015).
- R6.** Jaumann, T. *et al.* Lifetime vs. rate capability: Understanding the role of FEC and VC in high-energy Li-ion batteries with nano-silicon anodes. *Energy Storage Mater.* **6**, 26–35 (2017).
- R7.** Kim, K. *et al.* Understanding the thermal instability of fluoroethylene carbonate in LiPF₆-based electrolytes for lithium ion batteries. *Electrochim. Acta* **225**, 358–368 (2017).
- R8.** Schiele, A. *et al.* The critical role of fluoroethylene carbonate in the gassing of silicon anodes for lithium-ion batteries. *ACS Energy Lett.* **2**, 2228–2233 (2017).
- R9.** Schwenke, K. U., Solchenbach, S., Demeaux, J., Lucht, B. L. & Gasteiger, H. A. The impact of CO₂ evolved from VC and FEC during formation of graphite anodes in lithium-ion batteries. *J. Electrochem. Soc.* **166**, A2035–A2047 (2019).
- R10.** Deng, B. *et al.* Effects of charge cutoff potential on an electrolyte additive for LiNi_{0.6}Co_{0.2}Mn_{0.2}O₂ – mesocarbon microbead full cells. *Energy Technol.* **7**, 1800981 (2019).
- R11.** Zuo, X. *et al.* Effect of tris(trimethylsilyl)borate on the high voltage capacity retention of LiNi_{0.5}Co_{0.2}Mn_{0.3}O₂/graphite cells. *J. Power Sources* **229**, 308–312 (2013).
- R12.** Deng, B. *et al.* Investigating the influence of high temperatures on the cycling stability of a LiNi_{0.6}Co_{0.2}Mn_{0.2}O₂ cathode using an innovative electrolyte additive. *Electrochim. Acta* **236**, 61–71 (2017).
- R13.** Han, J.-G., Kim, K., Lee, Y. & Choi, N.-S. Scavenging materials to stabilize LiPF₆ - containing carbonate-based electrolytes for Li-ion batteries. *Adv. Mater.* **31**, 1804822 (2019).

We have newly added **References R1–R13** as references 18, 19, 22–28, 38–41 in the revised manuscript.

2. I was surprised to find out in the supplementary information that the employed anode contains only 3% Si. This fact should be clearly stated in the main part of the manuscript.

Response. We agree with your valuable comment. We have clearly described the content of Si of the Si-C anode based on the SNG/graphite composite on page 16 of the revised manuscript.

On page 17 (main text): The specific capacity and content of Si of the Si-C anode based on the SNG/graphite composite were 435.7 mAh g⁻¹ and 3 wt%, respectively.

3. More details such as supplier, composition, etc. should be given about the employed Si-C material.

Response. Thank you for your valuable comments. We have included the supplier and composition of electrodes in the revised manuscript.

On page 17 (main text): The Si-C anode was composed of 37.4 wt% Si nanolayer-embedded graphite (SNG with 7 wt% Si, SJ Advanced Materials), 58.6 wt% graphite (LA1, Shanshan (China)), 1 wt% carbon black (Super C65, Imerys Graphite & Carbon), and 3 wt% binding material (2 wt% styrene-butadiene rubber (BM-400B, Zeon) + 1 wt% carboxymethyl cellulose (MAC350H, Nippon Paper Group)) in distilled water and coated onto Cu foil (10 μm). The SNG was fabricated using a chemical vapor deposition (CVD) process according to literature⁵⁶.

Reference

R14. Ko, M. *et al.* Scalable synthesis of silicon-nanolayer-embedded graphite for high-energy lithium-ion batteries. *Nat. Energy* **1**, 16113 (2016).

Reference R14 was newly added as reference 56 in the revised manuscript.

4. The cell formation protocol also has to be disclosed in full detail including number of cycles at each rate, voltage range, use of constant voltage step, temperature etc.

Response. We thank the reviewer for the valuable comment regarding the cell formation protocol. We clearly state the number of cycles at each rate, voltage range, use of constant voltage step and temperature for precycling (1 time), standard cycles (2 times: 1 time at C/5 and 1 time at C/2), and cycle tests in the revised manuscript.

On page 18 (main text): Precycling for the formation of the SEI and CEI was performed at C/5 once. The cells were charged up to 4.3 V at C/5 followed by a constant voltage (CV) phase with a C/20 current cutoff; then, they were discharged to 2.5 V at 25 °C. Standard cycles with C/5 and C/2 for 1 time each were performed between 4.3 V and 2.5 V at 25 °C before subsequent cycling. A C/20 current cutoff was applied to finish the CV condition of the charge process. The GITT experiment was performed after two standard cycles (C/5 rate and C/2 rate once each). The cells were charged up to 4.3 V at C/5 for 5 min and then were left to

rest for 30 min to attain equilibrium voltage. A cycle test was performed without a CV condition at 1C at both 25 and 45 °C ($1C = 2.7 \text{ mA cm}^{-2}$).

5. How effectively does the proposed electrolyte stabilize electrodes with a higher Si content? The authors should compare the cycling stability of Si-rich anodes, e.g. anodes only containing the Si-C active material, using the multi-additive electrolyte to that using the FEC electrolyte.

Response. Thank you for your valuable comments. In our study, Si nanolayer-embedded graphite (SNG) with 7 wt% Si was blended with graphite to prepare the Si-C anode with 3 wt% Si (420.5 mAh g^{-1}). To confirm the effect of the proposed electrolyte on the cycling stability of full cells with Si-rich anodes, we evaluated the cycle performance of the NCM811/Si-C full cell containing the SNG with 7 wt% Si (520 mAh g^{-1}) as an anode (**Figure R1 and R2**). Clearly, a better capacity retention (77.0%) of the NCM811/Si-C full cell containing the SNG with 7 wt% Si (520 mAh g^{-1}) was achieved by using VC+DMVC-OCF₃+DMVC-OTMS after 400 cycles at 25 °C, delivering a high discharge capacity of 126.9 mAh g^{-1} and a high Coulombic efficiency of 99.7% (**Figure R1**).

These results have been added as **Supplementary Fig. 23** in the revised Supplementary Information and are discussed in the revised manuscript.

Figure R1. Cycle performance (a) and Coulombic efficiency (b) of Si-C anodes based on SNG with 7 wt% Si (520 mAh g^{-1}) coupled with NCM811 cathodes at a 1C rate and 25 °C.

Figure R2. Voltage profiles of Si-C anodes based on SNG with 7 wt% Si (520 mAh g^{-1}) coupled with NCM811 cathodes at 1C and 25 °C with the FEC containing electrolyte (a), VC containing electrolyte (b), and VC + DMVC-OCF₃ + DMVC-OTMS containing electrolyte (c) at the 1st, 100th, 200th, 300th, and 400th cycles.

On page 9 (main text): Further, VC+DMVC-OCF₃+DMVC-OTMS led to better cycling stability of full cells containing Si-C anodes with a higher Si content of 7 wt% than those of FEC and VC-added electrolytes (Supplementary Fig. 23).

6. To understand whether the proposed additive combination is superior in stabilizing Si-based anodes compared to FEC, half-cell tests vs. Li metal should be conducted for the Si-C anode in the proposed electrolyte and in the FEC-based electrolytes.

Response. Thank you for your valuable comments. To understand the effect of the proposed additive combination on the Si-C anode, a cycle test of Li/Si-C half-cells was performed. As shown in **Figure R3**, a capacity fading of the Li/Si-C half-cell after 20 cycles was observed in VC + DMVC-OCF₃ + DMVC-OTMS. This phenomenon occurs because DMVC-OCF₃ and DMVC-OTMS—which have a greater tendency for reduction than EC, VC, and FEC—undesirably react with the Li metal electrode in the half-cell (**Figure R4**). Thus, DMVC-OCF₃ and DMVC-OTMS could not contribute to the formation of a stable SEI on the Si-C anode and did not achieve good cycling stability of the Si-C anode in the half-cell.

Figure R3. Cycle performance (a) and Coulombic efficiency (b) of Li/Si-C half-cells at 1C and 25 °C after five formation cycles at C/5.

Figure R4. LUMO energy levels of EC, FEC, VC, DMVC-OCF₃, and DMVC-OTMS. Note that the isovalue of the orbital is 0.02 $e/\text{\AA}^3$.

Negative effect of VC on Li metal

Previously, it was demonstrated that the surface stability of the Li metal is assured by the formation of LiF-rich SEI, which suppresses the undesired reaction of the electrolyte with Li, and FEC is very effective to construct LiF-rich SEI on the Li metal^{R15}. More importantly, VC, which is regarded as an effective SEI former for graphite anodes, caused severe capacity fading of Li/LiNi_{0.6}Co_{0.2}Mn_{0.2}O₂ cells, as shown in **Figure R5**^{R16}.

Figure R5. (The result of the Liu group). Cycle performance of Li/LiNi_{0.6}Co_{0.2}Mn_{0.2}O₂ pouch cells with 1M LiPF₆ in EC/EMC (30:70 by weight) + 2 wt% VC with an electrolyte/capacity ratio of 3.0 g Ah⁻¹ (a), electrolyte/capacity ratio of 6.0 g Ah⁻¹ (b), and electrolyte/capacity ratio of 3.0 g Ah⁻¹, which got the refilling of fresh electrolyte into the cell (blue arrow) (c)^{R16}.

From previous results obtained by the Wang and Liu groups^{R15,R16}, we deduce that FEC-free electrolytes experience severe capacity fading in Li/cathode and Li/anode half-cells because of the interfacial instability of the Li metal. As the proposed electrolyte system does not contain FEC, which effectively produces LiF-rich SEI on the Li metal, we could confirm that the degradation of Li metal inhibits the elucidation of the beneficial effects of VC+DMVC-OCF₃+DMVC-OTMS on the Si-C anode and NCM811 cathode in the half-cell configuration with Li metal.

We would like to mention that the proposed electrolyte system is not intended for Li metal batteries but high-energy Li-ion batteries.

References

- R15.** Fan, X. *et al.* Non-flammable electrolyte enables Li-metal batteries with aggressive cathode chemistries. *Nat. Nanotechnol.* **13**, 715–722 (2018).
R16. Niu, C. *et al.* High-energy lithium metal pouch cells with limited anode swelling and long stable cycles. *Nat. Energy* **4**, 551–559 (2019).

Elucidation of unwanted reaction of VC + DMVC-OCF₃ + DMVC-OTMS with Li metal

As DMVC-OCF₃ and DMVC-OTMS have lower LUMO energies than VC and FEC, they are likely to decompose reductively when contacting Li metal in the half-cell configuration. This unwanted decomposition of DMVC-OCF₃ and DMVC-OTMS at the Li metal may hamper the formation of high-quality SEI on the Si-C anode in the Li/Si-C half-cell.

To elucidate the unwanted reaction between VC+DMVC-OCF₃+DMVC-OTMS and the Li metal electrode, the chemical structure of the surface of Li metal electrodes retrieved from Li/Si-C half-cells aged in different electrolytes for 20 h was investigated by X-ray photoelectron spectroscopy (XPS). The intensity of Li₂CO₃ (289.7 eV), C-O-C (533 eV), C=O (531.6 eV), and Li-O (529.5 eV) of the Li metal electrode aged in VC + DMVC-OCF₃ + DMVC-OTMS significantly increased, whereas the intensity of the LiF (684.7 eV) peak dramatically decreased (**Figure R6 and R7**). This result implies that DMVC-OCF₃ and DMVC-OTMS severely react with Li metal to form not LiF but oxygen-rich SEI species.

Figure R6. C 1s, O 1s, and F 1s XPS spectra of Li metal electrodes retrieved from Li/Si-C half-cells aged in different electrolytes at 25 °C for 20 h.

Figure R7. Composition of SEI on Li metal electrodes retrieved from Li/Si-C half-cells aged in different electrolytes at 25 °C for 20 h.

This undesired reaction of DMVC-OCF₃ and DMVC-OTMS with Li metal consumes the optimal content of DMVC-OCF₃ and DMVC-OTMS required for the protection of the Si-C anode (**Figure R8**); thus, the cycle performance of Si-C anodes in half-cells was inferior compared to that of FEC.

Figure R8. Reduction reaction of VC, DMVC-OCF₃, and DMVC-OTMS at the Li metal electrode in the Li/Si-C half-cell (a) and at the Si-C anode in the NCM811/Si-C full cell (b).

The parasitic reaction of DMVC-OCF₃ with the Li metal in the Li/Si-C half-cell was confirmed by ¹⁹F NMR measurement (**Figure R9**). Clearly, the signal corresponding to DMVC-OCF₃ was significantly reduced under no electric field. This is likely because DMVC-OCF₃ was consumed by the reaction with Li metal in Li/Si-C half-cell during aging for 20 h.

Figure R9. ^{19}F NMR spectra of a VC+DMVC-OCF₃+DMVC-OTMS electrolyte stored in a Li/Si-C half-cell and NCM811/Si-C full cell for 20 h. Peak areas corresponding to $-\text{OCF}_3$ of DMVC-OCF₃ were calculated relative to the peak area of an internal standard (C₆F₆).

To compare the reactivity of FEC, VC, and DMVC-OCF₃ toward the Li metal without an electric field, the Li metal was stored in each compound at 25 °C for 1 day. Li metal in contact with neat DMVC-OCF₃ was covered with byproducts generated by the undesired reaction between DMVC-OCF₃ and the Li metal (**Figure R10**). This result appears to be caused by the high reactivity of DMVC-OCF₃ toward the Li metal electrode because of the high electron affinity of DMVC-OCF₃ with relatively lower LUMO energy compared to those of VC and FEC.

Figure R10. Li metal in contact with FEC (a), VC (b), and DMVC-OCF₃ (c) at 25 °C for 1 day to compare their reaction tendencies without an electric field.

To understand the decomposition reaction of DMVC-OCF₃ and DMVC-OTMS on the Li metal surface, DFT calculations were carried out on the adsorption energies of the additives (DMVC-OCF₃, DMVC-OTMS, FEC, VC) on (001) surface of Li metal, which is the most

stable surface thermodynamically (**Figures R11 and R12**)^{R17}. The adsorptions of DMVC-OTMS and DMVC-OCF₃ on the (001) surface of the Li metal were found to be stronger than those of FEC and VC. Note that a more negative adsorption energy indicates stronger adsorption. This demonstrates that DMVC-OTMS and DMVC-OCF₃ are preferentially adsorbed on the surface of the Li metal. Furthermore, the LUMO energy levels of additives decreased after the adsorption on the Li metal surface. The decrements of LUMO energies of DMVC-OTMS and DMVC-OCF₃ were larger than those of FEC and VC, implying that DMVC-OTMS and DMVC-OCF₃ are likely to suffer reductive decomposition more aggressively after adsorption on the Li metal surface (**Figure R13**). Clearly, it is difficult to elucidate the beneficial effects of VC + DMVC-OCF₃ + DMVC-OTMS on the Si-C anode and NCM811 cathode in the half-cell configuration with the Li metal, as shown in **Table R1**.

Table R1. Comparison of working mechanism of FEC, VC and VC+DMVC-OCF₃+DMVC-OTMS in the half-cell configuration with the Li metal

	FEC	VC	VC + DMVC-OCF ₃ + DMVC-OTMS
Adsorption on Li metal	Relatively weak	Relatively weak	Stronger (DMVC-OCF ₃ , DMVC-OTMS)
LUMO energy	Low	Low	Very low (DMVC-OCF ₃ , DMVC-OTMS)
Li metal surface in contact with each electrolyte	LiF-rich SEI (from LiPF ₆)	LiF-rich SEI (from FEC)	LiF-less SEI
Li damage	Fair	Less	Severe

Reference

- R17.** Brennan, M. D., Breedon, M., Best, A. S., Morishita, T. & Spencer, M. J. S. Surface reactions of ethylene carbonate and propylene carbonate on the Li(001) surface. *Electrochim. Acta* **243**, 320–330 (2017).

Figure R11. Model systems of DMVC-OCF₃ (a), DMVC-OTMS (b), FEC (c), and VC (d) on the (001) surface of Li metal. Various types of additive configuration were considered. The surface of the Li metal consists of seven layers, which are the optimized number of layers in the (001) surface of the Li metal^{R18}, and the 5 × 4 supercell along the *x*- and *y*-axis. The two bottom layers are constrained. The vacuum spacing of model systems exceeds 20 Å.

Figure R12. Adsorption energies of various configurations of DMVC-OCF₃, DMVC-OTMS, FEC or VC on the Li (001) surface. The number labels below the bars indicate each adsorption configuration presented in **Figure R11**.

Figure R13. Changes in the LUMO energy levels of additives with adsorption on the Li metal surface. The dotted and solid lines indicate the LUMO energy levels of additives before and after the adsorption on the Li metal surface, respectively. The isovalue of the orbital is $0.02 e/\text{\AA}^3$.

These results have been added as **Supplementary Figs. 27-29, 33-36** in the revised

Supplementary Information and are discussed in the revised manuscript.

On page 10 (main text): The proposed electrolyte system underwent undesired decomposition at the Li metal in half-cell configuration because of the stronger adsorption and high reactivity of DMVC-OCF₃ and DMVC-OTMS toward the Li metal (Supplementary Figs. 27-32). Therefore, high-quality SEI by VC+DMVC-OCF₃+DMVC-OTMS was not formed on the Si-C anode in Li/Si-C half-cell, and the cathode-electrolyte interface was not maintained stably in the Li/NCM811 half-cell because of parasitic reactions between the DMVC-OCF₃ and Li metal (Supplementary Figs. 33-36).

The computational results have been added as **Supplementary Fig. 30-32** in the revised Supplementary Information and are discussed in the revised manuscript and Supplementary Information.

On page 10 (main text): The proposed electrolyte system underwent undesired decomposition at the Li metal in half-cell configuration because of the stronger adsorption and high reactivity of DMVC-OCF₃ and DMVC-OTMS toward the Li metal (Supplementary Figs. 27-32).

On page 19 (main text): In this study, we investigated the LUMO energy levels, reaction mechanisms, and adsorption energies using DFT calculations.

On page 4 (Supplementary Information): DFT calculations were performed to investigate the LUMO energy levels and reaction mechanisms. Beck's three-parameter hybrid functional combined with the Lee-Yang-Parr correlation (B3LYP) functional was used for the exchange-correlation energy^{2,3}.

On page 5 (Supplementary Information): For the calculations of the adsorption energies of additives (DMVC-OCF₃, DMVC-OTMS, FEC, VC) on the Li (001) surface, the generalized gradient approximation (GGA) with the Perdew-Burke-Ernzerhof (PBE) functional¹¹ was used for the exchange-correlation energy. The orbital cutoff and smearing value were set to 5.1 Å and 0.005 Ha, respectively. The 4×5×1 k-point was used with Monkhorst-Pack grid¹².

References

- R18.** Budi, A. *et al.* Study of the initial stage of solid electrolyte interphase formation upon chemical reaction of lithium metal and *N*-methyl-*N*-propyl-pyrrolidinium-bis(fluorosulfonyl)imide. *J. Phys. Chem. C* **116**, 19789–19797 (2012).

- R19.** Perdew, J. P., Burke, K. & Ernzerhof, M. Generalized gradient approximation made simple. *Phys. Rev. Lett.* **77**, 3865–3868 (1996).
- R20.** Monkhorst, H. J. & Pack, J. D. Special points for Brillouin-zone integrations. *Phys. Rev. B* **13**, 5188–5192 (1976).

Lewis acid (e.g. PF_5) promoted defluorination of FEC to give HF in LiPF_6 -based electrolytes is an unfavorable pathway in Li-ion batteries, which consequently produce excess HF in the cell^{R21-R23}, as depicted in **Figure R14**. This reaction also leads to transition metal leaching from the cathode at elevated temperatures; the dissolved transition metal cations move toward the lithiated anode when the full cell is stored at 60 °C. Then, these are deposited on the anode surface by taking the electrons from the charged anode. This electron consumption via transition metal deposition on the lithiated anode results in the reduction of the reversible capacity of the full cell^{R7,R24}. Indeed, the open circuit voltage (OCV) of a full cell with FEC drastically decreased compared to VC and VC + DMVC-OCF₃ + DMVC-OTMS, and its capacity retention was significantly reduced after 30-day storage at 60 °C (**Figure R15**). This result suggests that FEC-derived SEI and residual FEC, which was not consumed before the storage experiment, are thermally unstable and do not restrain the self-discharge of a full cell. Because of these critical issues raised by FEC, we would be reluctant to introduce FEC into Li-ion batteries, which should meet several performance requirements including high-temperature storage performance.

Figure R14. Defluorination of FEC by Lewis acids, such as PF_5 , in LiPF_6 -based electrolytes.

	OCV drop (V)	Capacity retention (%) after 30 days at 60 °C
No additive	0.43	44.0
FEC	0.25	60.7
VC	0.15	91.3
VC + DMVC-OCF ₃ + DMVC-OTMS	0.17	86.5

Figure R15. OCV changes (a) and capacity retention (b) of NCM811/Si-C full cells in fully charged stated (SOC 100) when stored for 30 days at 60 °C. NCM811/Si-C full cells with different electrolytes after precycling and standard cycles were charged up to 4.3V at a C/5 rate followed by CV with a C/20 current cutoff at 25 °C and then were stored at 60 °C. After 30 days at 60 °C, the capacity retention of the full cells was measured at a C/5 rate at 25 °C.

This result has been added as **Supplementary Fig. 37** in the revised Supplementary Information and is discussed in the revised manuscript.

On page 10 (main text): The open circuit voltage (OCV) drop of the fully charged NCM811/Si-C full cell with FEC was severe compared with VC and VC + DMVC-OCF₃ + DMVC-OTMS, and its capacity retention was significantly reduced to 60.7% after storing for 30 days at 60 °C (Supplementary Fig. 37). This result suggests that FEC-derived SEI and residual FEC, which is not consumed before storage experiment, are thermally unstable and do not restrain the self-discharge of a full cell. This is because of the undesirable defluorination of FEC, producing HF and acid compounds which promote transition metal ion dissolution from the cathode²⁶. The dissolved transition metal ions are then deposited on

the anode surface by taking the electrons from the charged anode, and the electron loss of anode inevitably causes a decrease in OCV and the capacity of charged full cells during storage at elevated temperatures.

References

- R21.** Shin, H., Park, J., Sastry, A. M. & Lu, W. Effects of fluoroethylene carbonate (FEC) on anode and cathode interfaces at elevated temperatures. *J. Electrochem. Soc.* **162**, A1683–A1692 (2015).
- R22.** Xu, C. *et al.* Unraveling and mitigating the storage instability of fluoroethylene carbonate-containing LiPF₆ electrolytes to stabilize lithium metal anodes for high-temperature rechargeable batteries. *ACS Appl. Energy Mater.* **2**, 4925–4935 (2019).
- R23.** Hernández, G. *et al.* Elimination of fluorination: The influence of fluorine-free electrolytes on the performance of LiNi_{1/3}Mn_{1/3}Co_{1/3}O₂/silicon–graphite Li-ion battery cells. *ACS Sustain. Chem. Eng.* **8**, 10041–10052 (2020).
- R24.** Jin, Y., Zhu, B., Lu, Z., Liu, N. & Zhu, J. Challenges and recent progress in the development of Si anodes for lithium-ion battery. *Adv. Energy Mater.* **7**, 1700715 (2017).

7. Half-cell tests should also be conducted for the NCM811 cathode to understand the influence of the novel additives on the stabilization of the cathode compared to VC and FEC.

Response. Thank you for your valuable comments. We evaluated the cycling stability of Li/NCM811 half-cells. The high reactivity and facile adsorption of DMVC-OCF₃ toward the Li metal electrode cause the creation of uncontrolled SEI composed of oxygen-rich and LiF-less species, which fatally damage the Li metal, as illustrated in **Figure R16**. Similar to the negative effect of VC + DMVC-OCF₃ + DMVC-OTMS on the cycle stability of Li/Si-C half-cells, VC + DMVC-OCF₃ + DMVC-OTMS did not properly protect the NCM811 cathode in the Li/NCM811 half-cells. To understand the failure mechanism of the Li/NCM811 half-cells, the NCM811 cathode retrieved from Li/NCM811 half-cells was recycled in a newly assembled half-cell with new Li metal, new electrolyte, and new separator. The capacity of the half-cell with VC + DMVC-OCF₃ + DMVC-OTMS was recovered to 178.1 mAh g⁻¹, while the reassembled half-cell with FEC was recovered to 174.7 mAh g⁻¹ (see the 51st cycle of **Figure R17**). From this result, we could confirm that the half-cell test using Li metal for elucidating the beneficial effect of VC + DMVC-OCF₃ + DMVC-OTMS on the NCM811 cathode is inappropriate.

Figure R16. Schematic illustration showing HF scavenging by TMS motif in VC + DMVC-OCF₃ + DMVC-OTMS-derived SEI on the Si-C anode in the NCM811/Si-C full cell (a) and undesirable reaction of DMVC-OCF₃ and DMVC-OTMS at the Li metal electrode and degradation of the NCM811 cathode with VC + DMVC-OCF₃ + DMVC-OTMS (b).

Figure R17. Cycle performance (a) and Coulombic efficiency (b) of Li/NCM811 half-cells at a 1C rate and 25 °C after five formation cycles at a C/5 rate.

8. The electrolytes should also be compared at a low rate such as C/5.

Response. Thank you for your valuable comments. We evaluated the cycle performance of NCM811/Si-C full cells at a C/5 rate and 25 °C. The full cell with VC + DMVC-OCF₃ +

DMVC-OTMS showed an improved capacity retention of 76.8% after 200 cycles, compared with FEC (73.9%) and VC (76.3%)-containing full cells (**Figure R18, R19, and Table R2**).

Figure R18. Cycle performance (a) and Coulombic efficiency (b) of NCM811/Si-C full cells with different electrolytes at a C/5 rate and 25 °C.

Figure R19. Voltage profiles of NCM811/Si-C full cells at a C/5 rate and 25 °C with FEC containing electrolyte (a), VC containing electrolyte (b), and VC + DMVC-OCF₃ + DMVC-OTMS containing electrolyte (c) at the 1st, 20th, 50th, 100th, 150th, and 200th cycles.

Table R2. Discharge capacity and capacity retention of NCM811/Si-C full cells after 200 cycles at a C/5 rate and 25 °C

	FEC	VC	VC + DMVC-OCF ₃ + DMVC-OTMS
Discharge capacity (mAh/g) @ 200th cycle	143.3	145.8	147.8

Retention (%)	73.9	76.3	76.8
---------------	------	------	------

This result has been added as **Supplementary Fig. 25** in the revised Supplementary Information and is discussed in the revised manuscript.

On page 9 (main text): In addition, NCM811/Si-C full cells with VC + DMVC-OCF₃ + DMVC-OTMS showed extremely stable cyclability during 1000 cycles with 80% depth of discharge (Supplementary Fig. 24) and the improved capacity retention after 200 cycles at a C/5 rate (Supplementary Fig. 25).

Reviewer #2 (Remarks to the Author):

The performance of VC + DMVC-OCF₃ + DMVC+ OTMS is only slightly better than the cells with FEC or VC. As there is no statistically convincing data to support that this new electrolyte is really making a significant difference, I don't have the confidence to support the acceptance of this article.

An inherent challenge of this work is that the study has treated this battery as a black box. The overall performance is evaluated without paying any attention to the performance of the cathode or anode alone. However, the argument of the positive impacts of this electrolyte is based on postulations pertaining to individual electrodes. This becomes a tautological argument. I hope the following suggestions are useful for the authors to improve their manuscript.

1. Please provide the potential profiles of the cathode and anode at different stages of the cycling.

Response. We thank the reviewer for the constructive suggestion. First, we would like to mention that the proposed electrolyte system in this study is not intended for Li metal batteries but high-energy Li-ion batteries. **Figure R20** shows the potential profiles of the cathode in the Li/NCM811 half-cells for the 1st, 3rd, 5th, and 10th. As shown in **Figure R21**, severe capacity fading of the Li/NCM811 half-cells was observed in VC + DMVC-OCF₃ + DMVC-OTMS after 20 cycles. This phenomenon occurs because DMVC-OCF₃ and DMVC-OTMS—which have a greater tendency for reduction than EC, VC, and FEC—undesirably

react with the Li metal electrode in the half-cell (**Figure R22**). Further, it is thought that the byproducts (oxygen-rich and LiF-less species) generated by the decomposition of DMVC-OCF₃ and DMVC-OTMS cause non-uniform plating of the Li ions supplied from the NCM811 cathode during the charging process, and severe deterioration of the Li metal shortens the cycle life of the half-cell.

Figure R20. Voltage profiles of Li/NCM811 half-cells at a 1C rate and 25 °C with the FEC containing electrolyte (a), VC containing electrolyte (b), and VC + DMVC-OCF₃ + DMVC-OTMS containing electrolyte (c) for the 1st, 3rd, 5th, and 10th cycles.

Figure R21. Cycle performance (a) and Coulombic efficiency (b) of Li/NCM811 half-cells at a 1C rate and 25 °C after five formation cycles at a C/5 rate.

Figure R22. Schematic illustration showing HF scavenging by TMS motif in VC + DMVC-OCF₃ + DMVC-OTMS-derived SEI on the Si-C anode in the NCM811/Si-C full cell (a) and undesirable reaction of DMVC-OCF₃ and DMVC-OTMS at the Li metal electrode and interfacial degradation of the NCM811 cathode with VC + DMVC-OCF₃ + DMVC-OTMS (b).

Negative effect of VC on Li metal

Previously, it was demonstrated that the surface stability of Li metal is assured by the formation of LiF-rich SEI, which suppress the undesired reaction of the electrolyte with the Li, and FEC is very effective to construct LiF-rich SEI on the Li metal^{R25}. More importantly, VC, which is regarded as an effective SEI former for graphite anodes, caused severe capacity fading of the Li/LiNi_{0.6}Co_{0.2}Mn_{0.2}O₂ cells, as shown in **Figure R23**^{R26}. From previous results obtained by the Wang and Liu groups^{R25,R26}, we deduce that FEC-free electrolytes experience severe capacity fading in the Li/cathode and Li/anode half-cells because of the interfacial instability of the Li metal. As the proposed electrolyte system does not contain FEC, which effectively produces LiF-rich SEI on the Li metal, we could confirm that the degradation of Li metal inhibits the elucidation of the beneficial effects of VC + DMVC-OCF₃ + DMVC-OTMS on the Si-C anode and NCM811 cathode in the half-cell configuration with the Li metal.

Figure R23 (The result of the Liu group). Cycle performance of Li/LiNi_{0.6}Co_{0.2}Mn_{0.2}O₂ pouch cells with 1M LiPF₆ in EC/EMC (30:70 by weight) + 2 wt% VC with an electrolyte/capacity ratio of 3.0 g Ah⁻¹ (a), electrolyte/capacity ratio of 6.0 g Ah⁻¹ (b), and electrolyte/capacity ratio of 3.0 g Ah⁻¹, which got the refilling of fresh electrolyte into the cell (blue arrow) (c)^{R26}.

References

- R25.** Fan, X. *et al.* Non-flammable electrolyte enables Li-metal batteries with aggressive cathode chemistries. *Nat. Nanotechnol.* **13**, 715–722 (2018).
R26. Niu, C. *et al.* High-energy lithium metal pouch cells with limited anode swelling and long stable cycles. *Nat. Energy* **4**, 551–559 (2019).

To understand the decomposition reaction of DMVC-OCF₃ and DMVC-OTMS on the Li metal surface, DFT calculations were carried out on the adsorption energies of the additives (DMVC-OCF₃, DMVC-OTMS, FEC, VC) on the (001) surface of Li metal, which is the most stable surface thermodynamically (**Figures R24**)^{R27}. The adsorptions of DMVC-OTMS and DMVC-OCF₃ on the (001) surface of the Li metal were found to be stronger than those of FEC and VC. Note that a more negative adsorption energy indicates stronger adsorption. This demonstrates that DMVC-OTMS and DMVC-OCF₃ are preferentially adsorbed on the surface of the Li metal. Furthermore, the LUMO energy levels of additives decreased after the adsorption on the Li metal surface. The decrements of LUMO energies of DMVC-OTMS and DMVC-OCF₃ were larger than those of FEC and VC, implying that DMVC-OTMS and DMVC-OCF₃ are likely to suffer the reductive decomposition more aggressively after adsorption on the Li metal surface (**Figure R25**).

Reference

- R27.** Brennan, M. D., Breedon, M., Best, A. S., Morishita, T. & Spencer, M. J. S. Surface reactions of ethylene carbonate and propylene carbonate on the Li(001) surface. *Electrochim. Acta* **243**, 320–330 (2017).

Figure R24. Adsorption energies of various configurations of DMVC-OCF₃, DMVC-OTMS, FEC or VC on the Li (001) surface. The number labels below the bars indicate each adsorption configuration presented in **Figure R11**.

Figure R25. Changes in the LUMO energy levels of additives with adsorption on the Li metal surface. The dotted and solid lines indicate the LUMO energy levels of additives before and after the adsorption on the Li metal surface, respectively. The isovalue of the orbital is

$0.02 e/\text{\AA}^3$.

The computational results have been added as **Supplementary Fig. 30-32** in the revised Supplementary Information and are discussed in the revised manuscript.

Moreover, DMVC-OCF₃ and DMVC-OTMS did not contribute to the formation of a stable SEI on the Si-C anode, and thereby good cycling stability of the Si-C anode could not be achieved in the half-cell (**Figure R26**).

Figure R26. Cycle performance (a) and Coulombic efficiency (b) of Li/Si-C half-cells at a 1C rate and 25 °C after five formation cycles at a C/5 rate.

Figure R27. Voltage profiles of Li/Si-C half-cells at a 1C rate and 25 °C with the FEC-containing electrolyte (a), VC-containing electrolyte (b), and VC + DMVC-OCF₃ + DMVC-OTMS containing electrolyte (c) at the 1st, 3rd, 5th and, 10th cycles. Voltage profiles of Li/Si-C half-cells at a 1C rate and 25 °C with different electrolytes at the 1st cycle (d).

As seen in **Figure R27**, the Li/Si-C half-cell with VC + DMVC-OCF₃ + DMVC-OTMS showed lower overpotential than the VC-containing half-cell during the 1st charge and discharge processes. However, because of undesired reaction between VC + DMVC-OCF₃ + DMVC-OTMS and the Li metal in half-cells, the capacity of the Li/Si-C half-cell faded after 20 cycles. The positive impact of VC + DMVC-OCF₃ + DMVC-OTMS on the Si-C anode could be confirmed through a comparison of the cycling stability between VC and VC + DMVC-OCF₃ + DMVC-OTMS.

To elucidate the unwanted reaction between VC + DMVC-OCF₃ + DMVC-OTMS and the Li metal electrode, the chemical structure of the surface of Li metal electrodes retrieved from the Li/Si-C half-cells aged in different electrolytes for 20 h was investigated by X-ray photoelectron spectroscopy (XPS). The intensity of Li₂CO₃ (289.7 eV), C-O-C (533 eV), C=O (531.6 eV), and Li-O (529.5 eV) of the Li metal electrode aged in VC + DMVC-OCF₃ + DMVC-OTMS significantly increased, whereas the intensity of the LiF (684.7 eV) peak dramatically decreased (**Figure R28 and R29**). This result implies that DMVC-OCF₃ and DMVC-OTMS severely react with the Li metal to form not LiF but oxygen-rich SEI species.

Figure R28. C 1s, O 1s, and F 1s XPS spectra of Li metal electrodes retrieved from Li/Si-C half-cells aged in different electrolytes for 20 h.

Figure R29. Composition of SEI composition on Li metal electrodes retrieved from Li/Si-C half-cells aged in different electrolytes 25 °C for 20 h.

This undesired reaction of DMVC-OCF₃ and DMVC-OTMS with the Li metal consumes the optimal content of DMVC-OCF₃ and DMVC-OTMS required for the protection of the Si-C anode (**Figure R30**); thus, cycle performance of the Si-C anodes in half-cells was inferior compared to that of FEC.

Figure R30. Reduction reaction of VC, DMVC-OCF₃ and DMVC-OTMS at the Li metal electrode in the Li/Si-C half-cell **(a)** and at the Si-C anode in the NCM811/Si-C full cell **(b)**.

Clearly, it is difficult to elucidate the beneficial effects of VC + DMVC-OCF₃ + DMVC-OTMS on the Si-C anode and NCM811 cathode in the half-cell configuration with the Li metal, as shown in **Table R3**.

Table R3. Comparison of working mechanism of FEC, VC, and VC + DMVC-OCF₃ + DMVC-OTMS in the half-cell configuration with the Li metal

	FEC	VC	VC + DMVC-OCF ₃ + DMVC-OTMS
Adsorption on Li metal	Relatively weak	Relatively weak	Stronger (DMVC-OCF ₃ , DMVC-OTMS)
LUMO energy	Low	Low	Very low (DMVC-OCF ₃ , DMVC-OTMS)
Li metal surface in contact with each electrolyte	LiF-rich SEI (from LiPF ₆)	LiF-rich SEI (from FEC)	LiF-less SEI
Li damage	Fair	Less	Severe

2. Particularly, I hope to see the profiles for the failing cycles at the end of the cycling.

Response. Thank you for your valuable comments. **Figure R31** and **Figure R32d-f** show the corresponding charge-discharge voltage curves at the 1st, 300th, 350th, and 400th cycles to observe the failing cycles at the end of the cycling. The NCM811/Si-C full cells with VC + DMVC-OCF₃ + DMVC-OTMS showed relatively small overpotential, which is mostly associated with the kinetics of the electrochemical reaction of the Si-C anode and NCM811 cathode, upon repeated cycling. It is plausible that VC + DMVC-OCF₃ + DMVC-OTMS helps in the formation of stable interface structures, allowing reversible lithiation and delithiation of both electrodes. Clearly, VC delivered reduced reversible capacity with large overpotential during cycling. This result implies that VC generates a resistive interfacial layer that impedes charge transport at the Si-C anode and NCM811 cathode.

Figure R31. (Supplementary Fig. 22. in the revised Supplementary Information). Voltage profiles of NCM811/Si-C full cells at a 1C rate and 25 °C with different electrolytes. No additive (a), FEC (b), VC (c), VC + DMVC-OCF₃ (d), and VC + DMVC-OCF₃ + DMVC-OTMS electrolytes (e) for the 1st, 300th, 350th, and 400th cycles. Coulombic efficiency of NCM811/Si-C full cells with different electrolytes at a 1C rate and 25 °C (f).

The voltage profiles of NCM811/Si-C full cells with FEC, VC and VC + DMVC-OCF₃ + DMVC-OTMS have been added as **Fig. 4d-f** in the revised manuscript.

Figure R32 (Fig. 4 in the revised manuscript). **Electrochemical performance of synthesized functional additives and fast charging capability.** **a**, dQ/dV graph of NCM811/Si-C full cells. (No additive: 1.15 M LiPF₆ in EC/EMC (3/7, v/v)) **b**, Charge and discharge GITT profiles and IR drop of NCM811/Si-C full cells. **c**, Cycle performance of NCM811/graphite full cells at 1C and 25 °C. **d-f**, Voltage profiles of NCM811/Si-C full cells at 1C and 25 °C with FEC containing electrolyte (**d**), VC containing electrolyte (**e**), and VC + DMVC-OCF₃ + DMVC-OTMS containing electrolyte (**f**) at the 1st, 300th, 350th, and 400th cycles. **g**, Charge rate capability of NCM811/Si-C full cells at a 1C discharge rate. **h**, Fast charging (1C and 3C) cycle performance of NCM811/Si-C full cells at a 1C discharge rate at 25 °C. **i**, XRD patterns and photographs of Si-C anodes charged (lithiated) at a 5C rate.

3. What about the cycling stability of both anode and cathode in half-cells?

Response. As shown in **Figures R33 and R34**, the capacity fading of the Li/Si-C and Li/NCM811 half-cells appeared for VC + DMVC-OCF₃ + DMVC-OTMS after 20 cycles. This phenomenon occurs because DMVC-OCF₃ and DMVC-OTMS—which have a greater tendency for reduction than EC, VC, and FEC—undesirably react with the Li metal electrode in the half-cell (**Figure R25**). The unwanted reaction between VC + DMVC-OCF₃ + DMVC-OTMS and the Li metal electrode was confirmed through an XPS study for the surface of the Li metal electrodes after contacting VC + DMVC-OCF₃ + DMVC-OTMS for 20 h at 25 °C (**Figure R28**).

Figure R33. Cycle performance (a) and Coulombic efficiency (b) of Li/Si-C half-cells at a 1C rate and 25 °C after five formation cycles at a C/5 rate.

Figure R34. Cycle performance (a) and Coulombic efficiency (b) of Li/NCM811 half-cells at a 1C rate and 25 °C after five formation cycles at a C/5 rate.

4. What is the specific capacity of the cathode and anode in half-cells?

Response. Thank you for your valuable comments. The specific charge and discharge capacities of the NCM811 cathode with VC + DMVC-OCF₃ + DMVC-OTMS at precycling were 226.1 mAh g⁻¹ and 201.4 mAh g⁻¹, respectively. The specific charge and discharge capacities of the Li/Si-C half-cell with VC + DMVC-OCF₃ + DMVC-OTMS at the formation cycle were 420.5 mAh g⁻¹ and 462.4 mAh g⁻¹, respectively. The voltage profiles of the Li/NCM811 half-cells and Li/Si-C half-cells are shown in **Figure R35**.

Figure R35. Voltage profiles of Li/NCM811 half-cells (**a**), and Li/Si-C half-cells (**b**) during precycling at a C/10 rate. The formation cycle of the Li/NCM811 half-cells at a C/10 rate for 1 time were performed between 4.35 V and 3.0 V at 25 °C. A C/20 current cutoff was applied to finish the CV condition of the charge process. The formation cycle of the Li/Si-C half-cells at a C/10 rate for 1 time were performed between 0.005 V and 1.0 V at 25 °C. A C/100 current cutoff was applied to finish the CV condition of the charge process.

VC + DMVC-OCF₃ + DMVC-OTMS showed a higher discharge capacity of 201.4 mAh g⁻¹ in the Li/NCM811 half-cell compared with FEC (200.1 mAh g⁻¹) and VC (198.8 mAh g⁻¹), as shown in **Table R4**. VC + DMVC-OCF₃ + DMVC-OTMS showed similar discharge capacity (420.5 mAh g⁻¹) with an initial Coulombic efficiency of 90.9% with FEC (420.2 mAh g⁻¹, 91.0%) in the Li/Si-C half-cells. Most importantly, the Li/NCM811 and Li/Si-C half-cells with VC + DMVC-OCF₃ + DMVC-OTMS showed lower voltage loss owing to small overpotential at precycling, compared with those of FEC and VC. The degradation of the Li metal by the undesired reaction between DMVC-OCF₃ and the Li metal in half-cells may be not significant at precycling.

Table R4. Specific capacity and initial Coulombic efficiency of the Li/NCM811 half-cells with different electrolytes during precycling at a C/10 rate

	Charge capacity (mAh g ⁻¹)	Discharge capacity (mAh g ⁻¹)	Initial Coulombic efficiency (%)
FEC	223.5	200.1	89.5
VC	222.3	198.8	89.4
VC + DMVC-OCF ₃ + DMVC-OTMS	226.1	201.4	89.1

Table R5. Specific capacity and initial Coulombic efficiency of the Li/Si-C half-cells with different electrolytes during precycling at a C/10 rate

	Charge capacity (mAh g ⁻¹)	Discharge capacity (mAh g ⁻¹)	Initial Coulombic efficiency (%)
FEC	461.7	420.2	91.0
VC	447.0	402.1	90.0
VC + DMVC-OCF ₃ + DMVC-OTMS	462.4	420.5	90.9

5. What is the N/P ratio?

Response. Thank you for your valuable comment. The full cell was designed with an N/P ratio of 1.3. The specific charge capacity and discharge capacity of Li/NCM811 half-cell were 226.1 mAh g⁻¹ and 201.4 mAh g⁻¹, respectively, with VC + DMVC-OCF₃ + DMVC-OTMS at the formation cycle with a C/10 rate and voltage range between 4.35 V and 3.0 V at 25 °C. The specific charge, discharge capacity, and irreversible capacities of the Li/Si-C half-cell were 420.5 mAh g⁻¹, 462.4 mAh g⁻¹ and 40.9 mAh g⁻¹, respectively, with VC + DMVC-OCF₃ + DMVC-OTMS at the formation cycle with a C/10 rate and voltage range between 0.005 V and 1.0 V at 25 °C. This point was clearly described in the revised manuscript.

On page 18 (main text): 2032 coin-type full cells were fabricated in an argon-filled glove box, and an N/P ratio of 1.3 was determined using Eq. 1.

$$\frac{(\text{Discharge capacity of anode}) \times (\text{Mass of anode})}{(\text{Discharge capacity of cathode}) \times (\text{Mass of cathode}) - (\text{Irreversible capacity of anode}) \times (\text{Mass of anode})} = \frac{(420.5 \text{ mAh/g}) \times (7.5 \text{ mg/cm}^2)}{(201.4 \text{ mAh/g}) \times (13.5 \text{ mg/cm}^2) - (40.9 \text{ mAh/g}) \times (7.5 \text{ mg/cm}^2)} = 1.3 \quad (1)$$

6. The font of the words in Figure 1c looks strange. Please fix it.

Response. Thank you for your valuable comment. We corrected the font and size of the words to improve the readability of Fig. 1c in the revised manuscript.

7. Figure 3a, the LUMO energy level of EC is positive. What is the reference?

Response. We appreciate the valuable comment regarding the calculated result of the LUMO energy level of EC. In our DFT calculation, we used the B3LYP functional as the exchange-correlation energy to calculate the LUMO energy levels of the electrolytes and additives. Generally, the B3LYP functional is well-known as a hybrid functional of the DFT method for calculating electronic properties, such as orbital energy levels. The B3LYP functional is considered to be more accurate than the GGA-PBE functional, which is commonly used for the exchange-correlation energy. There exist some cases where the LUMO level are negative, but many previous studies showed positive LUMO energy levels of EC using the B3LYP functional (**R28-R33**), as in this case.

Reference

- R28.** Bhatt, M. D., Cho, M. & Cho, K. Interaction of Li⁺ ions with ethylene carbonate (EC): Density functional theory calculations. *Appl. Surf. Sci.* **257**, 1463–1468 (2010).
- R29.** Bhatt, M. D. & O'Dwyer, C. The role of carbonate and sulfite additives in propylene carbonate-based electrolytes on the formation of SEI layers at graphitic Li-ion battery anodes. *J. Electrochem. Soc.* **161**, A1415–A1421 (2014).
- R30.** Seo, D. M. *et al.* Role of mixed solvation and ion pairing in the solution structure of lithium ion battery electrolytes. *J. Phys. Chem. C* **119**, 14038–14046 (2015).
- R31.** Zhang, L., Huang, Y., Fan, H. & Wang, H. Flame-retardant electrolyte solution for dual-ion batteries. *ACS Appl. Energy Mater.* **2**, 1363–1370 (2019).
- R32.** Yue, H. *et al.* Boron additive passivated carbonate electrolytes for stable cycling of 5 V lithium-metal batteries. *J. Mater. Chem. A* **7**, 594–602 (2019).
- R33.** Feng, D. *et al.* Mixed lithium salts electrolyte improves the high-temperature performance of nickel-rich based lithium-ion batteries. *J. Electrochem. Soc.* **167**, 110544 (2020).

8. Figure 7e, why with VC + DMVC-OCF₃ + DMVC+ OTMS, the thickness of the Si-C anode is much bigger?

Response. As seen in **Figure R36**, Fig. 7e presents the thickness of the Si-C anode cycled with VC during 400 cycles at 25 °C. The thickness change of the Si-C anode with VC + DMVC-OCF₃ + DMVC+ OTMS is displayed in Fig. 7f of the revised manuscript.

Figure R36. SEM image of a pristine Si-C anode (a) and Si-C anodes retrieved from NCM811/Si-C full cells after 400 cycles at 25 °C with VC (b) or VC + DMVC-OCF₃ + DMVC-OTMS (c), cross-sectional images of the pristine Si-C anode (d), and Si-C anodes from NCM811/Si-C full cells after 400 cycles at 25 °C with VC (e), or VC + DMVC-OCF₃ + DMVC-OTMS (f). g-i, EDS mapping in TEM of the pristine Si-C anode (g) and Si-C anodes after 400 cycles with VC (h) or VC + DMVC-OCF₃ + DMVC-OTMS (i).

Reviewer #3 (Remarks to the Author):

In this work, two functional VC derivatives of DMVC-OCF₃ and DMVC-OTMS are synthesized and studied as the electrolyte additive for a Si/C-NMC811 cell. The results are meaningful, and might be suitable for publication after revising and clarifying the questions below:

1. Bromine in DMVC-Br is relatively active, which should be easily hydrolyzed. Why the synthesis of DMVC-OH do not use the direct hydrolysis of DMVC-Br, instead, a two-step synthesis route?

Response. We thank the reviewer for the positive evaluation and constructive suggestion. As the reviewer mentioned, allyl bromide (DMVC-Br) obtained from DMVC through radical bromination is reactive toward nucleophilic substitution reactions. Direct hydrolysis conditions were attempted; however, H₂O under neutral conditions (**Table R6**, entries 1,3; see also **Figure R37** for TLC) revealed low conversion. Enhancing the reaction temperature (60 °C) caused the decomposition of DMVC-Br, which is associated with the competitive electrophilicity of carbonate carbon of DMVC-Br (entries 2,4). The addition of NaOH did not facilitate DMVC-OH formation (entries 5,6). Thus, the hydroxymethyl (carbinol) moiety of DMVC-OH was introduced through a two-step sequence (**Scheme R1**) using the readily hydrolysable formic ester intermediate with enhanced reaction yields^{R34}.

Table R6. Attempted direct hydrolysis conditions of DMVC-Br

entry	attempted hydrolysis conditions	isolated yield (%): DMVC-OH
1	H ₂ O as a solvent at rt	<1%
2	H ₂ O as a solvent at 60 °C	2%
3	H ₂ O:MeCN = 1:1 as solvent at rt	<1%
4	H ₂ O:MeCN = 1:1 as solvent at 60 °C	<1%
5	NaOH (1 equiv) in H ₂ O: MeCN = 1:1 at rt	<1%

Figure R37. Attempted hydrolysis of DMVC-Br and TLC analysis.

Scheme R1. Synthesis of DMVC-OH through formic ester intermediate

The formic ester intermediate (DMVC-OCOCH₃) was prepared from ally bromide (DMVC-Br) by using formic acid and triethylamine (TEA) in acetonitrile. Then, DMVC-OCOCH₃ was readily hydrolyzed to DMVC-OH (**Scheme R1**). The synthetic details are given below.

Synthetic details of DMVC-Br: DMVC (5.0 g, 1.0 equiv) was treated with *N*-bromosuccinimide (NBS, 8.2 g, 1.1 equiv) and azobisisobutyronitrile (AIBN, 98 μL) in 1,2-DCE (30 mL). The resulting mixture was stirred at 100 °C for 4 h. The title compound was yielded through flash chromatography (ethyl acetate/*n*-hexane, 7:1) and afforded the title

product as a yellow oil (8.3 g, 98%); ¹H NMR (400 MHz, CDCl₃) δ 4.18 (s, 2H), 2.13 (s, 3H).

Synthetic details of DMVC-OH: Triethylamine (18.1 mL, 3.0 equiv) was dropped into a solution of DMVC-Br (8.3 g, 1.0 equiv) and formic acid (4.9 mL, 3.0 equiv) in acetonitrile (80 mL) while keeping the temperature under 0 °C. The resulting mixture was stirred at room temperature for 2 h. After the concentration of the reaction mixture, the residue was diluted with EtOAc. The organic phase was washed with water and brine and was concentrated to give DMVC-OCOH (5.0 g). This intermediate was then dissolved in methanol (75 mL), and 37% HCl (0.25 mL) was added. After stirring for 5 h at room temperature, the reaction mixture was concentrated. The title compound was yielded through flash chromatography (ethyl acetate/n-hexane, 1:1) as a yellow oil (4.1 g, 73%); ¹H NMR (400 MHz, CDCl₃) δ 4.42 (s, 2H), 2.14 (s, 3H).

Reference

R34. Alpegiani, M., Zarini, F., & Perrone, E. On the preparation of 4-hydroxymethyl-5-methyl-1,3-dioxol-2-one. *Synth. Commun.* **22**, 1277–1282 (1992).

This result has been added as **Supplementary Fig. 1, Supplementary Fig. 2, and Supplementary Table 1** in the revised Supplementary Information and is discussed in the revised manuscript and Supplementary Information.

On page 5 (main text): DMVC-OH as a synthetic platform was prepared in 72% isolated yield in 3 steps, namely, radical bromination, formate ester generation, and hydrolysis (Fig. 1a; see also the Supplementary Methods and Supplementary Fig. 1 of the Supplementary Information)⁴². The synthetic route involving the formation of the readily hydrolysable formate ester intermediate was selected owing to the higher yield compared to the direct hydrolysis conditions (Supplementary Table 2 and Supplementary Fig. 2).

On page 2-4 (Supplementary Information): Synthesis of DMVC-OH by direct hydrolysis of DMVC-Br: the allyl bromide (DMVC-Br) obtained from DMVC through radical bromination is reactive toward nucleophilic substitution reactions. Direct hydrolysis conditions were attempted; however, H₂O under neutral conditions revealed low conversion (Supplementary Table 2, entries 1,3; see also Supplementary Fig. 2 for TLC). Enhancing the reaction temperature (60 °C) caused the decomposition of DMVC-Br, which is associated with the competitive electrophilicity of carbonate carbon of DMVC-Br (entries 2,4). The addition of

NaOH did not facilitate DMVC-OH formation (entries 5,6). Thus, the hydroxymethyl (carbinol) moiety of DMVC-OH was introduced through a two-step sequence (Supplementary Fig. 3) using the readily hydrolysable formate ester intermediate with enhanced reaction yields¹. The formate ester intermediate (DMVC-OCOH) was prepared from allyl bromide (DMVC-Br) by using formic acid and triethylamine (TEA) in acetonitrile. Then, DMVC-OCOH was readily hydrolyzed to DMVC-OH (Supplementary Fig. 1).

Synthetic details of DMVC-Br: DMVC (5.0 g, 1.0 equiv) was treated with *N*-bromosuccinimide (NBS, 8.2 g, 1.1 equiv) and azobisisobutyronitrile (AIBN, 98 μ L) in 1,2-dichloroethane (DCE, 30 mL). The resulting mixture was stirred at 100 °C for 4 h. The title compound was yielded through flash chromatography (ethyl acetate/*n*-hexane, 7:1) and afforded the title product as a yellow oil (8.3 g, 98%); ¹H NMR (400 MHz, CDCl₃) δ 4.18 (s, 2H), 2.13 (s, 3H).

Synthetic details of DMVC-OH: Triethylamine (18.1 mL, 3.0 equiv) was dropped into a solution of DMVC-Br (8.3 g, 1.0 equiv) and formic acid (4.9 mL, 3.0 equiv) in acetonitrile (80 mL) while keeping the temperature under 0 °C. The resulting mixture was stirred at room temperature for 2 h. After concentration of the reaction mixture, the residue was diluted with EtOAc. The organic phase was washed with water and brine and concentrated to give DMVC-OCOH (5.0 g). This intermediate was then dissolved in methanol (75 mL), and 37% HCl (0.25 mL) was added. After stirring for 5 h at room temperature, the reaction mixture was concentrated. The title compound was yielded through flash chromatography (ethyl acetate/*n*-hexane, 1:1) as a yellow oil (4.1 g, 73%); ¹H NMR (400 MHz, CDCl₃) δ 4.42 (s, 2H), 2.14 (s, 3H).

2. In Fig. 1b, the O=CF₂ product is a gas and highly reactive to all of the SEI, CEI, solvents, and electrode materials (Si and NMC811), it cannot be stably present in the cell. Please propose a reasonable final product.

Response. We thank the reviewer for raising this point. Fluorophosgene (COF₂) is a highly electrophilic intermediate bearing electronegative fluorides (**Scheme R2**)^{R35}. Conversion of COF₂ with nucleophilic substances may furnish the corresponding carbon dioxide or organic carbonate derivatives^{R36–R38}. As suggested, we have included reasonable secondary products in **Fig. 1** of the revised manuscript (see **Figure R38**).

Scheme R2. Conversion of COF₂ to secondary products

This result has been added as **Supplementary Fig. 4** in the revised Supplementary Information and is discussed in the revised manuscript.

References

- R35.** Feng, P., Lee, K. N., Lee, J. W., Zhan, C. & Ngai, M. Y. Access to a new class of synthetic building blocks via trifluoromethoxylation of pyridines and pyrimidines. *Chem. Sci.* **7**, 424–429 (2016).
- R36.** Farlow, M. W., Man, E. H. & Tullock, D. W. Carbonyl fluoride. *Inorganic Syntheses*. (Ed. Rochow, E. G.) **6**, 155–158 (1960).
- R37.** Avataneo, M., De Patta, U., Galimberti, M. & Marchionni, G. Synthesis of α,ω -dimethoxyfluoropolyethers: reaction mechanism and kinetics. *J. Fluorine Chem.* **126**, 631–637 (2005).
- R38.** Petzold, D., Nitschke, P., Brandl, F., Scheidler, V., Dick, B., Gschwind, R. M. & König, B. Visible-light-mediated liberation and in situ conversion of fluorophosgene. *Chem. Eur. J.* **25**, 361–366 (2019).

We have included reasonable secondary products of COF₂ in **Fig. 1** of the revised manuscript.

On page 6 (main text): Conversion of COF₂ with nucleophilic substances generated by the reductive decomposition of DMVC-OCF₃ may furnish the corresponding carbon dioxide or organic carbonate derivatives^{45–47} (Supplementary Fig. 4).

We have added **References R32-R35** as references 42, 45–47 in the revised manuscript.

Figure R38 (Fig. 2 in the revised manuscript). **Synthesis of functional VC derivatives and transformation of additives to form SEI on the Si-C anode.** **a**, Synthesis of DMVC-OTMS and DMVC-OCF₃. **b**, Electrochemical transformations of DMVC derivatives. SET = Single electron transfer; NBS = *N*-bromosuccinimide; AIBN = azobisisobutyronitrile; 1,2-

DCE = 1,2-dichloroethane; TEA = trimethylamine; MeCN = acetonitrile; EtOAc = ethyl acetate. c, Design of a deformable and stable SEI using VC, DMVC-OCF₃ and DMVC-OTMS on the Si-C anode.

3. Both DMVC-OCF₃ and DMVC-OTMS contain >C=C< vinyl groups that are unstable at high potentials. How they affect the performance of NCM811 cathode in addition to HF-scavenge?

Response. We appreciate the valuable comment regarding the stability of DMVC-OCF₃ and DMVC-OTMS with the >C=C< vinyl group at high potentials. To explore the oxidation stability of DMVC-OCF₃ and DMVC-OTMS, the leakage current of Li/NCM811 half-cells was monitored at a constant charging voltage of 4.35V vs. Li/Li⁺ for 3 h. Clearly, VC + DMVC-OCF₃ + DMVC-OTMS showed much reduced leakage currents, which indicates better oxidation stability of the electrolyte (**Figure R39**). This result suggests that the >C=C< vinyl group of DMVC-OCF₃ and DMVC-OTMS does not badly affect the performance of the NCM811 cathode at high potentials.

Figure R39. Leakage current of Li/NCM811 half-cells with different electrolytes at a constant voltage of 4.35 V vs. Li/Li⁺.

During cycling, conventional electrolytes with VC or FEC form resistive and fragile SEI on the Si-C anode, which impedes delithiation of the anode, and the extent of re-lithiation into the NCM811 cathode may be reduced (**Figure R40**). This phenomenon induces the capacity decay of a full cell and the undesirable potential increase of the NCM811 cathode to adjust the charge cutoff voltage of a full cell during further charge process. It is noteworthy that potential increase of the NCM811 cathode in a full cell provokes oxidative decomposition of

the electrolyte at the cathode, which leads to the undesired reduction of Ni^{4+} to lower oxidation states of Ni^{3+} and Ni^{2+} , resulting in the formation of the rock-salt phase. Contrarily, VC + DMVC-OCF₃ + DMVC-OTMS constructs the stable SEI on the Si-C anode, which promotes the reversible delithiation/lithiation of the anode. Thus, better capacity retention of NCM811/Si-C full cells could be achieved in VC + DMVC-OCF₃ + DMVC-OTMS.

Figure R40. Schematic representation for VC + DMVC-OCF₃ + DMVC-OTMS-derived SEI, which effectively inhibits the unwanted increment in the cathode potential by facilitating the lithiation/delithiation of the anode.

4. In Supplementary Fig. 3a, there are two dQ/dV peaks for DMVC-OCF₃, what reduction does the other peak at higher potential correspond to? Besides, DMVC-OTMS chart is missed in Supplementary Fig. 3b.

Response. Thank you for your valuable comment. As shown in the dQ/dV plot of **Figure R41a**, the potential scan was conducted from open circuit voltage to 0.005 V in Li/Si-C half-cell with DMVC-OCF₃. Prior to the insertion of Li ions into the Si-C anode, two pronounced reduction peaks (peak I and II of **Figure R41a**), associated with the reductive decomposition of DMVC-OCF₃, were observed. In the initial state, the reductive decomposition of DMVC-OCF₃ into the DMVC radical and OCF₃ anion by one-electron reduction occurred favorably because the LUMO energy level of the OCF₃ radical was much lower than those of the DMVC radical and decomposed DMVC-OCF₃ by C=C bond cleavage (Supplementary Fig. 7 of the revised Supplementary Information). Therefore, peak I of **Figure R41a** is attributable to the decomposition of DMVC-OCF₃ into the DMVC radical and OCF₃ anion (**Figure**

R41c). The second peak shown in **Figure R41a** is attributable to the reduction of the OCF_3 anion to form LiF because the LiF peak intensity drastically increased after lithiation to 0.45 V (**Figure R41a and b**).

Figure R41. dQ/dV plot of the Li/Si-C half-cell with DMVC-OCF₃ (**a**), XPS F 1s spectra of Si-C anodes at different voltages during first lithiation (**b**), and reductive decomposition mechanisms of DMVC-CF₃ during charging of precycling at 25 °C (**c**).

To distinguish the reduction peak of DMVC-OTMS in a combination of additives (VC + DMVC-OCF₃ + DMVC-OTMS), a DMVC-OTMS chart was added, as shown in **Figure R42b**. The data are reflected in Supplementary Fig. 6b of the revised Supplementary Information.

Figure R42. dQ/dV plots for the formation cycle of Li/Si-C half-cells with each synthesized additive (a) and a combination of additives (b).

This result has been added as **Figure S6b and S8** in the revised Supplementary Information and is discussed in the revised manuscript.

On page 7 (main text): The first lithiation of Li/Si-C half-cell also exhibits a reduction peak at 1.0 V vs. Li/Li⁺, which implies a one-electron reduction of DMVC-OCF₃ to form the DMVC radical and OCF₃ anion (Supplementary Fig. 8a and c). The second peak (see Supplementary Fig. 8a) is attributable to the reduction of the OCF₃ anion to form LiF, because the LiF peak intensity was drastically increased after lithiation to 0.45 V (Supplementary Fig. 8a and b).

5. In Fig. 4b, R_{sei} and R_{ct} are changed within the error ranges, especially the right two (they have almost the same R_{sei} and R_{ct}). It appears that the impedance results cannot explain so large differences in the capacity at 5C in Fig. 4d.

Response. Thank you for your valuable comment. As the reviewer mentioned, there was no significant difference in the impedance of NCM811/Si-C full cells after precycling. To confirm the effect of additives on the fast charging of NCM811/Si-C full cells, galvanostatic intermittent titration technique (GITT) experiments were performed (**Figure R43 and R44b**). Notably, the NCM811/Si-C full cell with VC + DMVC-OCF₃ + DMVC-OTMS showed reduced IR drop compared with VC and FEC. This reduced IR drop implies that VC +

DMVC-OCF₃ + DMVC-OTMS constructs less resistive interfacial layers on electrodes, allowing facile ion transport at high charge C-rates. Although the NCM811/Si-C full cell without additives showed relatively low IR drop at early cycling, the impedance of the full cell was drastically increased after cycling because of the continuous electrolyte decomposition at the anode (**Supplementary Fig. 19** in the revised Supplementary Information). This is because additive-free electrolyte forms EC-derived SEI, which nonuniformly covers the anode surface^{R39,R40}, and the anode with less coverage by the SEI has relatively low resistance. This non-uniform SEI does not prevent further electrolyte decomposition at the anode during cycling and causes continuous increment in the cell resistance.

Figure R43. Charge and discharge GITT profiles and IR drop of NCM811/Si-C full cells with different electrolytes after standard cycles (C/5 rate for 1 time and C/2 rate for 1 time).

The GITT result of NCM811/Si-C full cells has been added as **Fig. 4b** in the revised manuscript.

On page 8-9 (main text): Importantly, galvanostatic intermittent titration technique (GITT) experiments confirmed that the NCM811/Si-C full cell with VC + DMVC-OCF₃ + DMVC-OTMS exhibits reduced IR drop by less resistive interfacial layers compared with full cells with VC or FEC, allowing facile ion migration at high charge C-rates (Fig. 4b).

References

- R39.** Nguyen, C. C. & Lucht, B. L. Comparative study of fluoroethylene carbonate and vinylene carbonate for silicon anodes in lithium ion batteries. *J. Electrochem. Soc.* **161**, A1933–A1938 (2014).
- R40.** Abraham, D. P., Furczon, M. M., Kang, S. H., Dees, D. W. & Jansen, A. N. Effect of

electrolyte composition on initial cycling and impedance characteristics of lithium-ion cells. *J. Power Sources* **180**, 612–620 (2008).

Figure R44 (Fig. 4 in the manuscript). **Electrochemical performance of synthesized functional additives and fast charging capability.** **a**, dQ/dV graph of NCM811/Si-C full cells. (No additive: 1.15 M LiPF₆ in EC/EMC (3/7, v/v)) **b**, Charge and discharge GITT profiles and IR drop of NCM811/Si-C full cells. **c**, Cycle performance of NCM811/graphite full cells at 1C and 25 °C. **d-f**, Voltage profiles of NCM811/Si-C full cells at a 1C rate and 25 °C with FEC containing electrolyte (**d**), VC containing electrolyte (**e**), and VC + DMVC-OCF₃ + DMVC-OTMS containing electrolyte (**f**) at the 1st, 300th, 350th, and 400th cycles. **g**, Charge rate capability of NCM811/Si-C full cells at a 1C discharge rate. **h**, Fast charging (1C

and 3C) cycle performance of NCM811/Si-C full cells at a 1C discharge rate at 25 °C. **i**, XRD patterns and photographs of Si-C anodes charged (lithiated) at a 5C rate.

6. More details are needed in experimental: What potential/voltage ranges were used for the measurements of NMC811 (2.7 mA h cm^{-2}) and anode (3.2 mA h cm^{-2}) loadings? Is the Si-C a mixture or a Si-coated C composite? If it is a Si-C mixture, why there are no gaps between the Si and graphite boundaries in the TEM images (Figs. 5a-c); What is the Si content in Si-C mixture/composite?

Response. We thank the reviewer for the valuable comments. The voltage range of the full cells with the NMC811 (2.7 mA h cm^{-2}) cathode and Si-C anode (3.2 mA h cm^{-2}) was between 4.3 V and 2.5 V. The Si-C anode was comprised of a Si nanolayer-embedded graphite (SNG) with 7 wt% Si and graphite. The SNG was fabricated using a chemical vapor deposition (CVD) process according to the literature^{R14}. Because high-purity silane (SiH_4) gas was utilized to obtain a homogeneous amorphous Si nanolayer on graphite, there was no gap between the graphite and Si nanolayer in **Figs. 5a-c**. We have clearly detailed the experimental conditions and Si content of the SNG/graphite composite anode in the revised manuscript.

On page 17 (main text): The Si-C anode was composed of 37.4 wt% Si nanolayer-embedded graphite (SNG with 7 wt% Si, SJ Advanced Materials), 58.6 wt% graphite (LA1, Shanshan (China)), 1 wt% carbon black (Super C65, Imerys Graphite & Carbon), and 3 wt% binding material (2 wt% styrene-butadiene rubber (BM-400B, Zeon) + 1 wt% carboxymethyl cellulose (MAC350H, Nippon Paper Group)) in distilled water and coated onto Cu foil (10 μm). The SNG was fabricated using a chemical vapor deposition (CVD) process according to the literature⁵⁶. The specific capacity and content of Si of the Si-C anode based on the SNG/graphite composite were 435.7 mAh g^{-1} and 3 wt%, respectively.

On page 18 (main text): Precycling for the formation of the SEI and CEI was performed at C/5 once. The cells were charged up to 4.3 V at C/5 followed by a constant voltage (CV) phase with a C/20 current cutoff; then, they were discharged to 2.5 V at 25 °C. Standard cycles with C/5 and C/2 for 1 time each were performed between 4.3 V and 2.5 V at 25 °C before subsequent cycling. A C/20 current cutoff was applied to finish the CV condition of the charge process. The GITT experiment was performed after two standard cycles (C/5 rate and C/2 rate 1 time each). The cells were charged up to 4.3 V at C/5 for 5 min and then were left

to rest for 30 min to attain equilibrium voltage. A cycle test was performed without a CV condition at 1C at both 25 and 45 °C (1C = 2.7 mA cm⁻²).

Reference

R14. Ko, M. *et al.* Scalable synthesis of silicon-nanolayer-embedded graphite for high-energy lithium-ion batteries. *Nat. Energy* **1**, 16113 (2016).

Reference R14 was newly added as reference 56 in the revised manuscript.

On page 26 (main text):

56. Ko, M. *et al.* Scalable synthesis of silicon-nanolayer-embedded graphite for high-energy lithium-ion batteries. *Nat. Energy* **1**, 16113 (2016).

Reviewer #1 (Remarks to the Author):

The authors prepared an extensive revision including multiple new experiments. In my opinion, the authors sufficiently addressed the comments of the reviewers and hence I recommend that the study is now accepted for publication.

Reviewer #3 (Remarks to the Author):

My questions and comments have been well addressed. If Figure R39 appears in the Supplementary information, the name and value of its vertical axis should be converted into current density (mA/cm² or μ A/cm²), otherwise, this revised manuscript can be accepted in its present form.

Sheng S. Zhang

Manuscript ID: NCOMMS-20-27266A

Manuscript Type: Article

Title: Replacing conventional battery electrolyte additives with dioxolone derivatives for high-energy-density lithium-ion batteries

Dear Editor and Reviewer,

We thank you for your thoughtful suggestions and insights. The manuscript has benefited from these insightful suggestions. Below, we present our point-by-point response to the remarks, along with our modifications (highlighted in light blue) in the revised manuscript.

Reviewer #3 (Remarks to the Author):

My questions and comments have been well addressed. If Figure R39 appears in the Supplementary information, the name and value of its vertical axis should be converted into current density (mA/cm^2 or $\mu\text{A}/\text{cm}^2$), otherwise, this revised manuscript can be accepted in its present form.

Response. We thank the reviewer for the valuable comment and favorable evaluation of our manuscript. As suggested, the y-axis legend of **Figure R1** (Figure R39 in the previous “Response to referees” file) has been changed to current density (mA/cm^2). **Figure R1** has been provided as **Supplementary Fig. 23** in the revised Supplementary Information and is discussed in the revised manuscript.

Figure R1. Leakage current density of Li/NCM811 half-cells with different electrolytes at a constant voltage of 4.35 V vs. Li/Li⁺.

On page 9 (main text): To determine the oxidation stability of DMVC-OCF₃ and DMVC-OTMS, the leakage current of Li/NCM811 half-cells was monitored at a constant charging voltage of 4.35 V vs. Li/Li⁺ for 3 h. Compared to FEC and VC, VC + DMVC-OCF₃ + DMVC-OTMS showed reduced leakage current, which indicates higher oxidation stability of the electrolyte. This result suggests that the presence of the C=C vinyl group of DMVC-OCF₃ and DMVC-OTMS does not negatively affect the performance of the NCM811 cathode at high potentials (Supplementary Fig. 23).